# Structural basis for the regulation of human 5,10-methylenetetrahydrofolate reductase by phosphorylation and S-adenosylmethionine inhibition

D. Sean Froese [1], Jolanta Kopec[2], Elzbieta Rembeza[2], Gustavo Arruda Bezerra[2], Anselm Erich Oberholzer[3], Terttu Suormala[1], Seraina Lutz[1], Rod Chalk[2], Oktawia Borkowska[2], Matthias R. Baumgartner [1] & Wyatt W. Yue [2]

The folate and methionine cycles are crucial for biosynthesis of lipids, nucleotides and proteins, and production of the methyl donor S-adenosylmethionine (SAM). 5,10-methylene-tetrahydrofolate reductase (MTHFR) represents a key regulatory connection between these cycles, generating 5-methyltetrahydrofolate for initiation of the methionine cycle, and undergoing allosteric inhibition by its end product SAM. Our 2.5 Å resolution crystal structure of human MTHFR reveals a unique architecture, appending the well-conserved catalytic TIM-barrel to a eukaryote-only SAM-binding domain. The latter domain of novel fold provides the predominant interface for MTHFR homo-dimerization, positioning the N-terminal serine-rich phosphorylation region near the C-terminal SAM-binding domain. This explains how MTHFR phosphorylation, identified on 11 N-terminal residues (16 in total), increases sensitivity to SAM binding and inhibition. Finally, we demonstrate that the 25-amino-acid inter-domain linker enables conformational plasticity and propose it to be a key mediator of SAM regulation. Together, these results provide insight into the molecular regulation of MTHFR.

[1] Division of Metabolism and Children's Research Center, University Children's Hospital, CH-8032 Zürich, Switzerland. [2] Structural Genomics Consortium, Nuffield Department of Clinical Medicine, University of Oxford, Oxford OX3 7DQ, UK. [3] Structural Biology Community Laenggasse (sbcl), CH-3000 Bern, Switzerland. These authors contributed equally: D. Sean Froese, Wyatt W. Yue. Correspondence and requests for materials should be addressed to D.S.F. (email: sean.froese@kispi.uzh.ch) or to W.W.Y. (email: wyatt.yue@sgc.ox.ac.uk)

In humans, the folate and methionine cycles both generate products essential to cellular survival. Folate, the major cellular carrier of single carbon units, is required for the synthesis of purines and thymidine monophosphate. Within the methionine cycle, the methylation of homocysteine to methionine by methionine synthase (EC 2.1.1.13) produces an essential amino acid which may be used for protein synthesis or, crucially, be further converted to S-adenosylmethionine (SAM), a vitally important donor for the methylation of DNA, RNA and proteins as well as the creation of numerous methylated compounds. These two cycles intersect at the enzyme 5,10-methylenetetrahydrofolate reductase (MTHFR; E.C. 1.5.1.20). MTHFR catalyzes the physiologically irreversible reduction of 5,10-methylene-tetrahydrofolate ($CH_2$-THF) to 5-methyl-tetrahydrofolate ($CH_3$-THF), a reaction requiring FAD as a cofactor and NADPH as an electron donor. Since the product $CH_3$-THF is exclusively used by methionine synthase, and only the demethylated form (THF) may be recycled back to the folate cycle, MTHFR commits THF-bound one-carbon units to the methionine cycle.

In accordance with this essential role, major and minor deficiencies of human MTHFR are the direct or indirect causes of human disease. Severe MTHFR deficiency (MIM #607093) is inherited in an autosomal recessive manner and is the most common inborn error of folate deficiency[1] with ~200 patients known[2]. To date, over 100 different clinically relevant mutations in MTHFR have been described, the majority of which are of the missense type ($n = 70$, >60%) and private[2]. Milder enzyme deficiencies, due to single nucleotide polymorphisms of the MTHFR gene, have been associated with various common disorders. The most studied of these is p.Ala222Val (c.665C>T in NM_001330358, commonly annotated as c.677C>T), identified as a risk factor for an overwhelming number of multifactorial disorders, including vascular diseases, neurological diseases, various cancers, diabetes and pregnancy loss (see e.g. review by Liew and Gupta[3]).

Human MTHFR is a 656 amino acid multi-domain protein (Fig. 1). The catalytic domain is conserved across evolution, and crystal structures of MTHFR from Escherichia (E.) coli[4–7] and Thermus thermophilus[8], in which the catalytic domain constitutes the entire sequence (Fig. 1), have been solved. These structures reveal the catalytic domain to form a $\beta_8\alpha_8$ (TIM) barrel and have uncovered residues critical for binding the cofactor FAD[4], the electron donor NADPH (NADH in bacteria[7]) and the product $CH_3$-THF[5–7]. The bacterial structures, together with activity assay of trypsin cleaved porcine MTHFR[9], indicate that the catalytic domain is sufficient for the entire catalytic cycle. Eukaryotic MTHFR orthologs additionally possess a C-terminal regulatory domain that is connected to the catalytic domain by a linker sequence (Fig. 1). This C-terminal domain is able to bind SAM, resulting in allosteric inhibition of enzymatic activity[10], an effect which is very slow[11] and can be reversed by binding to S-adenosylhomocysteine (SAH)[12,13], the demethylated form of SAM.

Human MTHFR further contains a 35 amino acid serine-rich region at the very N-terminus which is not found in MTHFR orthologs of bacteria, yeast or even lower animals (Fig. 1). This region has been identified to be multiply phosphorylated following heterologous expression in insect cells[14] and yeast[15], or following immunoprecipitation from human cancer cell lines[16]. Phosphorylation has been associated with moderately decreased catalytic activity[14–16] and increased total inhibition mediated by SAM[14]. Although phosphorylation mapping of this region has been thus far unsuccessful, scanning mutagenesis has revealed substitution of alanine for threonine at position 34 (p.Thr34Ala) to almost completely block phosphorylation[14,15], suggesting Thr34 is the priming position. The cellular relevance of this modification remains unclear, although one group has suggested that phosphorylation at Thr34 can be accomplished by CDK1/cyclin B1 (ref. [16]) and at Thr549 by polo-like kinase 1 (ref. [17]) whereby they posit a role in histone methylation and replication.

The repertoire of bacterial MTHFR structures to date does not provide any mechanistic insight into the enzymatic regulation by phosphorylation and SAM binding, because both features are absent in prokaryotes. To this end, we have combined structural, biophysical and biochemical data of human MTHFR to provide a molecular view of MTHFR function and regulation in higher eukaryotes. We have identified specific phosphorylation sites and demonstrate a distinct relationship between phosphorylation and SAM inhibition. Further, using our 2.5 Å resolution crystal structure of the almost full-length human protein, we reveal that the regulatory domain utilizes a unique topology to bind SAH/SAM and transmit a catalytic inhibition signal, likely by long-range conformational change through the linker region.

## Results

**Phosphorylated residue identification by mass spectrometry.** To examine the phosphorylation status of human (Hs)MTHFR, we generated full-length recombinant human MTHFR ($HsMTHFR_{1–656}$) by baculovirus expression in Sf9 cells. Mass spectrometry-based phosphorylation mapping (with 92% coverage) identified 16 separate phosphorylation sites in $HsMTHFR_{1–656}$ following purification from Sf9 cells (called here as purified) (Fig. 2a). All phosphorylation sites were considered to have partial occupancy, since no residues were phosphorylated in every tryptic peptide analysed (Supplementary Fig. 1). Of these, 11 phosphorylated amino acids (Ser9, Ser10, Ser18, Ser20, Ser21, Ser23, Ser25, Ser26, Ser29, Ser30, Thr34) were within the N-terminal serine-rich region, including the putative phosphorylation determining residue Thr34 (Fig. 2a). Additionally, we found phosphorylation of three further amino acids in the catalytic domain (Tyr90, Thr94, Ser103) and two in the regulatory domain (Ser394, Thr451). Up to ten phosphorylation sites were identified to be occupied simultaneously, whereby treatment with calf intestine alkaline phosphatase (CIP) resulted in removal of 9 (Fig. 2b) or 10 (Fig. 2c) phosphate groups, as identified by

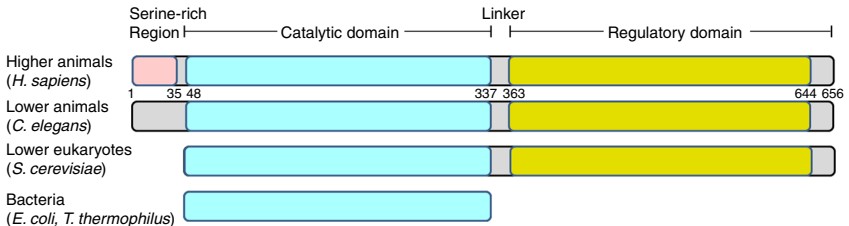

**Fig. 1** Schematic representation of MTHFR. Domain organization of MTHFR orthologs across evolution. Numbers given represent approximate amino acid boundaries in human MTHFR corresponding to NP_005948. In brackets is shown representative species within each category

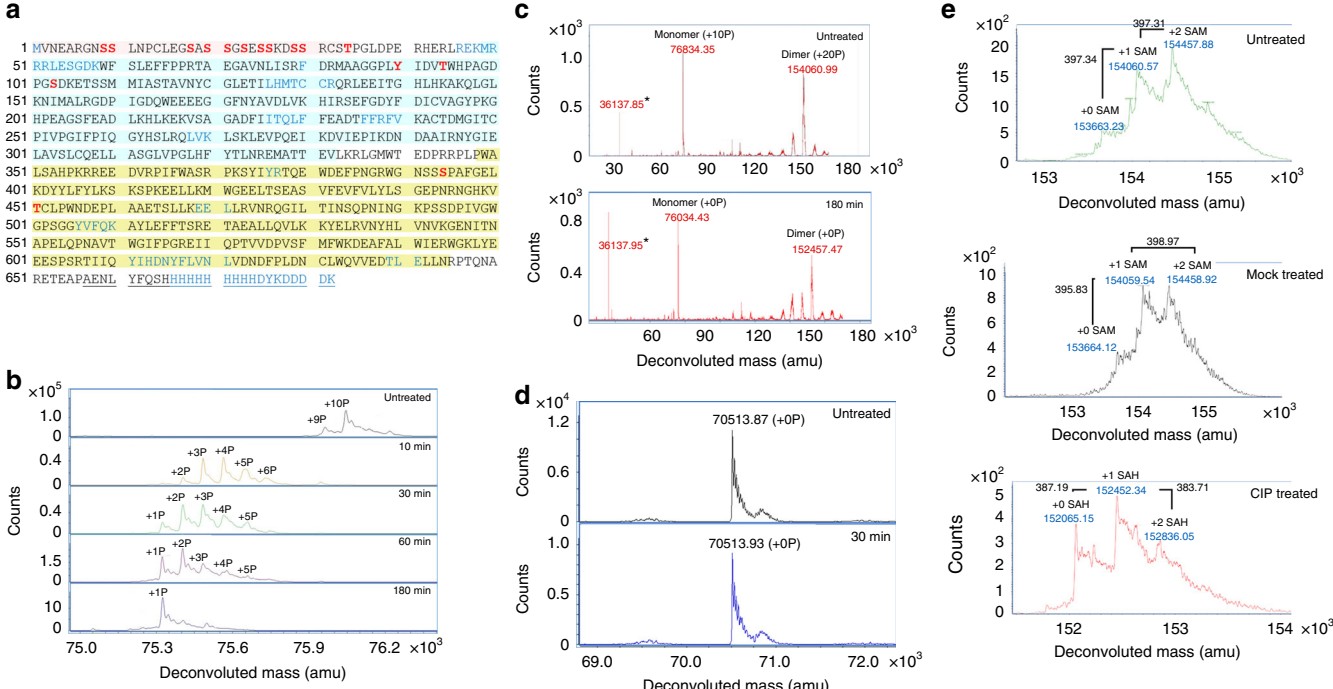

**Fig. 2** Phosphorylation status of $HsMTHFR_{1-656}$ and $HsMTHFR_{38-644}$. **a** Phosphorylation mapping of $HsMTHFR_{1-656}$. The protein sequence is given as amino acids in single letter code, including the C-terminal His/flag-tag (underlined). Black font represents amino acids identified by the mass spectrometer (covered), blue font represents amino acids not identified (non-covered), red font represents phosphorylated amino acids. Domains are coloured as in Fig. 1. **b** Dephosphorylation of $HsMTHFR_{1-656}$ following treatment with CIP. Treatment time at 37 °C is given. Large number above peaks represents number of phosphate groups attached. Proteins were analysed by denaturing mass spectrometry. amu, atomic mass units. **c** Native mass spectrometry analysis of $HsMTHFR_{1-656}$ before and after treatment with CIP. Upper panel: as purified (untreated) protein. Monomer represents protein bound to 1 FAD plus 10 phosphate groups (expected mass: 76831.16 amu); dimer represents protein bound to 2 FADs and 1 SAM plus 20 phosphate groups (expected mass: 154060.74 amu). Lower panel: protein following 180 min treatment with CIP. Monomer represents protein bound to 1 FAD (expected mass: 76031.16 amu); dimer represents protein bound to 2 FADs and 1 SAH (expected mass: 152446.74 amu). Expected sizes: protein without first methionine, 75245.6 amu; FAD, 785.56 amu; SAM, 398.44 amu; SAH, 384.42 amu, phosphate, 80.00 amu. * indicates a truncated protein representing amino acids 353–663 (expected mass: 36136.6 amu). **d** $HsMTHFR_{38-644}$ before and after treatment with CIP. Treatment time at 37 °C is given. Protein was analysed by denaturing mass spectrometry. **e** Native mass spectrometry of $HsMTHFR_{1-656}$ identifying sequential binding of SAM or SAH. Graphs represent areas zoomed in on dimeric protein. Upper panel: As purified (untreated) protein. Middle panel: control protein (heated in assay buffer for 180 min without CIP). Bottom panel: treated protein (heated in assay buffer for 180 min with CIP). Expected size of protein with 2 FAD bound and 20 phosphates: 153662.3. Expected size of protein with 2 FAD bound and 0 phosphates: 152062.32. Expected size of SAM: 398.44, SAH: 384.41

denaturing and native mass spectrometry, respectively. To examine the importance of the N-terminal serine-rich region to global protein phosphorylation, we produced recombinant $HsMTHFR_{38-644}$, which removes the N-terminal 37 amino acids, including the entire serine-rich region (Fig. 1) as well as the poorly conserved C-terminal 12 amino acids predicted to be of high disorder (Supplementary Fig. 2). As purified $HsMTHFR_{38-644}$ was not found to be phosphorylated by phosphorylation mapping (Supplementary Fig. 3a), or native mass spectrometry (Supplementary Fig. 3b), and treatment with CIP did not alter the protein molecular mass (Fig. 2d). Therefore, the primary determinant of $HsMTHFR$ phosphorylation resides within the N-terminus.

**Phosphorylation does not alter MTHFR kinetic parameters.** Phosphorylation has been described to alter MTHFR kinetics, resulting in moderately decreased catalytic activity as measured by the NADPH-menadione oxidoreductase assay[14,16]. To investigate this more thoroughly, we used a very sensitive high-performance liquid chromatography (HPLC)-based activity assay which monitors the full enzymatic reaction in the physiological direction and allows determination of kinetic values[18]. Overall, we found similar kinetic values for $HsMTHFR_{1-656}$ and $HsMTHFR_{38-644}$ (Table 1, Supplementary Fig. 4). As described in

Table 1, compared to non-phosphorylated $HsMTHFR_{38-644}$, phosphorylated $HsMTHFR_{1-656}$ had only slightly decreased specific activity ($34.0 \pm 1.3$ vs. $30.8 \pm 1.5$ μmol $min^{-1}$ $mg^{-1}$) and $k_{cat}$ values ($51.4 \pm 4.2$ vs. $40.7 \pm 3.2$ $s^{-1}$), suggesting turnover was not meaningfully reduced. These values are comparable but higher than previous determinations from recombinant $HsMTHFR$ (12.4 μmol $min^{-1}$ $mg^{-1}$ (ref. [13])) and purified porcine MTHFR (19.4 μmol $min^{-1}$ $mg^{-1}$ (ref. [19])). Importantly, CIP-treated $HsMTHFR_{1-656}$ (dephosphorylated) retained nearly identical activity values to mock-treated $HsMTHFR_{1-656}$ (i.e. assay without addition of CIP; phosphorylated) (Table 1), confirming that phosphorylation does not decrease enzymatic turnover. Likewise, Michaelis–Menten constants ($K_M$) for the substrate $CH_2$-THF (range: 21.3–25.5 μM) and electron donor NADPH (range: 23.5–35.5 μM) were very similar for all four proteins (Table 1), and comparable to that of MTHFR from human fibroblast lysates ($CH_2$-THF: 26 μM; NADPH: 30 μM[18]). Thus, we conclude that our assay is sensitive and specific, and that phosphorylation does not significantly alter MTHFR kinetic properties.

Interestingly, we found no increase in the specific activity of MTHFR proteins following addition of exogenous FAD to the assay buffer (Table 1). Since FAD is required as cofactor for the MTHFR reaction, this suggests the cofactor was already bound to the as purified protein, presumably acquired during cellular

**Table 1 Kinetic characterization of *Hs*MTHFR**

| Protein | Specific activity (μmol min⁻¹ mg⁻¹) | | | Heat-stable activity (%)[a] | | | Apparent $K_M$ values (μM) | | | Inhibition (μM) |
|---|---|---|---|---|---|---|---|---|---|---|
| | ØFAD | +FAD | $k_{cat}$ (s⁻¹) | ØFAD | Assay+FAD | Pre-FAD | CH2-THF | NADPH | NADH | SAM $K_i$ (95% CI) |
| *Hs*MTHFR₁₋₆₅₆ | 30.5 ± 0.9 | 30.8 ± 1.5 | 40.7 ± 3.2 | 33.9 ± 3.1 | 39.8 ± 1.4 | 81.0 ± 3.8 | 22.4 ± 1.3 | 35.5 ± 2.4 | 3760 ± 410 | 2.7 (2.2–3.5) |
| *Hs*MTHFR₃₈₋₆₄₄ | 34.8 ± 1.2 | 34.0 ± 1.3 | 51.4 ± 4.2 | 40.7 ± 3.9 | 83.7 ± 2.8 | 101.8 ± 3.2 | 25.5 ± 1.7 | 23.5 ± 1.7 | 2160 ± 270 | 21 (19–23) |
| *Hs*MTHFR₁₋₆₅₆ (mock) | 33.6 ± 3.1 | 32.9 ± 3.7 | 40.4 ± 4.0 | 32.6 ± 2.0 | 40.0 ± 0.8 | 82.2 ± 2.8 | 22.5 ± 3.3 | 30.3 ± 4.0 | N.D. | 3.8 (3.4–4.3) |
| *Hs*MTHFR₁₋₆₅₆ (CIP) | 33.2 ± 1.7 | 34.4 ± 2.5 | 39.9 ± 4.2 | 36.7 ± 1.6 | 43.2 ± 1.6 | 81.6 ± 2.7 | 21.3 ± 1.8 | 28.7 ± 2.3 | N.D. | 6.4 (6.1–6.7) |

[a]Heat-stable activity was measured following incubation of the assay mixture for 5 min at 46 °C
ØFAD, FAD was not supplemented; +FAD, FAD was added to the assay buffer after heating; pre-FAD, FAD was added to the protein before heating; N.D., not determined
All values represent the results of at least three separate experiments and are given as ±S.D.

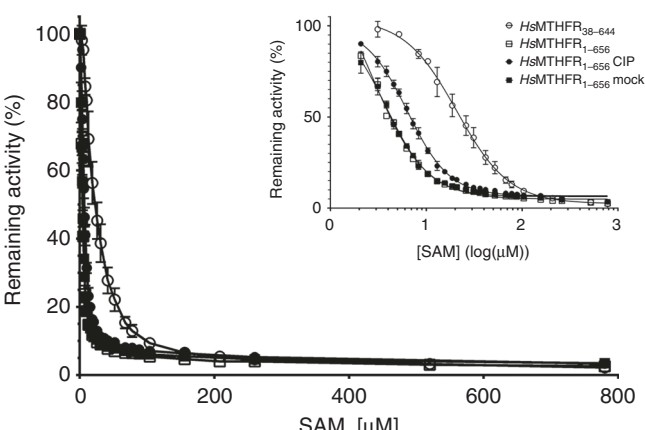

**Fig. 3** SAM inhibition of *Hs*MTHFR₁₋₆₅₆ and *Hs*MTHFR₃₈₋₆₄₄. Inhibition of MTHFR catalytic activity following pre-incubation with various concentrations of SAM. Remaining activity represents percentage of activity compared to MTHFR incubated without SAM. Inset: Replot of percent activity remaining against SAM concentration transformed by log10 to reveal differences between truncated (*Hs*MTHFR₃₈₋₆₄₄) and dephosphorylated full-length (*Hs*MTHFR₁₋₆₅₆ CIP) protein with phosphorylated full-length protein (*Hs*MTHFR₁₋₆₅₆; *Hs*MTHFR₁₋₆₅₆ mock). Inhibitory constants ($K_i$) for SAM calculated from this graph were calculated as described in Methods and are provided in Table 1. Each value represents the results of at least three separate experiments and is given as ±S.D

expression. This is consistent with native mass spectrometry, which identified monomeric and dimeric forms of as purified *Hs*MTHFR₁₋₆₅₆ which, in addition to phosphorylation, contained equivalent units of FAD (Fig. 2c). *Hs*MTHFR₃₈₋₆₄₄ also presented as monomeric and dimeric forms bound to equivalent units of FAD (Supplementary Fig. 3b), suggesting phosphorylation has no effect in this regard. Supplementation with FAD, however, helped rescue activity when provided either before, or to a lesser extent following, incubation of these proteins at 46 °C for 5 min (Table 1). Therefore, this cofactor, which is important for protein stability, may be lost under heat treatment. In our experiments, *Hs*MTHFR₁₋₆₅₆ was markedly more sensitive to heat inactivation than *Hs*MTHFR₃₈₋₆₄₄, but this heat sensitivity was not affected by the phosphorylation state of the protein, and therefore likely rather reflects overall protein stability.

**Phosphorylation increases SAM protection and sensitivity.** In addition to phosphorylation and FAD, native mass spectrometry identified the dimeric form of as purified *Hs*MTHFR₁₋₆₅₆ to contain 0, 1 or 2 units of SAM (Fig. 2e). Like FAD, SAM was likely acquired during cellular expression. However, following

CIP treatment, the SAM bound to MTHFR was found to degrade to SAH, a chemical transition which did not occur during mock treatment of the protein (Fig. 2e). Correspondingly, as purified dimeric *Hs*MTHFR₃₈₋₆₄₄, which is not phosphorylated, was found to be bound to 0, 1 or 2 units of SAH, but not SAM (Supplementary Fig. 3c). Thus, phosphorylated MTHFR appeared to protect thermally unstable SAM from degradation to SAH, while the non-phosphorylated protein was unable to perform this function.

Phosphorylation has been identified to affect the maximum degree of inhibition of MTHFR by SAM, whereby phosphorylated protein was found to be maximally ~80% inhibited, while phosphatase treated protein was maximally ~60% inhibited[14]. At high concentrations of SAM (>200 μM), we were able to inhibit all recombinant *Hs*MTHFR proteins by over 95%, regardless of the phosphorylation state (Fig. 3). However, at low SAM concentrations we found phosphorylated *Hs*MTHFR₁₋₆₅₆ to be more sensitive to SAM inhibition than *Hs*MTHFR₃₈₋₆₄₄ and dephosphorylated *Hs*MTHFR₁₋₆₅₆ (Fig. 3). Further analysis revealed as purified and mock-treated *Hs*MTHFR₁₋₆₅₆ to have inhibition constants ($K_i$) of ~3 μM, while CIP-treated *Hs*MTHFR₁₋₆₅₆ was approximately two-fold less sensitive to SAM inhibition, and *Hs*MTHFR₃₈₋₆₄₄ seven-fold less sensitive (Fig. 3, inset; Table 1). Thus, although phosphorylation does not directly affect MTHFR enzymatic activity, it increases the protein's sensitivity to SAM inhibition.

**An extensive linker connects and interacts with both domains.** We determined the 2.5 Å resolution structure of *Hs*MTHFR₃₈₋₆₄₄ in complex with FAD and SAH by multiple-wavelength anomalous dispersion using the selenomethionine (SeMet) derivatized protein (Table 2). The identity of both ligands is guided by well-defined electron density (Supplementary Fig. 5), and in line with native mass spectrometry analysis for this construct (Supplementary Fig. 3B). The *Hs*MTHFR₃₈₋₆₄₄ protomer folds into two globular domains (Fig. 4a) to form an overall elongated molecule. As predicted from bacterial structures, the N-terminal catalytic domain (aa 40–337) consists of an 8α/8β TIM barrel, adorned with three extra α-helices (α8, α9 and α11) (Fig. 4a, Supplementary Fig. 6). The C-terminal regulatory domain (aa 363–644) makes up a novel fold of two five-stranded β-sheets arranged side-by-side in the core, flanked by a number of α-helices (Fig. 4a, Supplementary Fig. 6). The two domains do not contact each other directly, but instead are connected by an extended linker region encompassing aa 338–362 (Fig. 4a), with its amino acid sequence rich in Arg ($n = 4$), Pro (4), Glu (3) and Leu (3). Mediated by three β-turns, this linker makes multiple intricate contacts with both domains, and changes direction twice in traversing between the catalytic and regulatory domains.

The *Hs*MTHFR₃₈₋₆₄₄ structure allows the mapping of the 70 inherited missense mutations known to cause severe MTHFR

**Table 2 Data collection and refinement statistics**

| | HsMTHFR$_{38-644}$ Native | | | | ScMET12$_{1-301}$ |
|---|---|---|---|---|---|
| | **Native** | **SeMet (MAD)** | | | |
| **Data collection** | | | | | |
| Space group | P2$_1$2$_1$2$_1$ | P2$_1$2$_1$2$_1$ | | | P2$_1$2$_1$2 |
| Cell dimensions | | | | | |
| $a$, $b$, $c$ (Å) | 97.3, 127.9, 147.0 | 97.24, 127.26, 147.29 | | | 110.6, 54.5, 61.9 |
| $\alpha$, $\beta$, $\gamma$ (°) | 90.00, 90.00, 90.00 | 90.00, 90.00, 90.00 | | | 90.00, 90.00, 90.00 |
| | | **Peak** | **Inflection** | **Remote** | |
| Wavelength | 0.9281 | 1.00605 | 1.00916 | 0.99835 | 0.9282 |
| Resolution (Å) | 68.53–2.50 (2.64–2.50) | | 48.62–3.00 (3.11–3.00) | | 22.75–1.56 (1.60–1.56) |
| $R_{sym}$ or $R_{merge}$ | 8.6 (136.2) | 4(57.8) | 3.8(58.6) | 4.1(60.4) | 9.2 (149.8) |
| $I$ /$\sigma I$ | 12.8 (1.2) | 18.6(1.2) | 18.6(1.2) | 18.2(1.1) | 14.3 (1.3) |
| Completeness (%) | 100 (100) | 100(100) | 100(99.9) | 100(99.9) | 99.5 (99.1) |
| Redundancy | 7.5 (7.0) | 2.0(2.0) | 2.0(2.0) | 2.0(2.0) | 8.4 (8.6) |
| **Refinement** | | | | | |
| Resolution (Å) | 96.51–2.50 | | | | 22.75–1.56 |
| No. of reflections (total/unique) | 479,094/64,233 | | | | 448,825/53,602 |
| $R_{work}$/$R_{free}$ | 21.03/24.75 | | | | 15.97/19.93 |
| No. of atoms | | | | | |
| Protein | 8914 | | | | 2410 |
| Ligand/ion | 184 | | | | 53 |
| Water | 72 | | | | 258 |
| $B$-factors | | | | | |
| Protein | 92.29 | | | | 19.78 |
| Ligand/ion | 98.25 | | | | 23.38 |
| Water | 64.09 | | | | 34.65 |
| R.m.s deviations | | | | | |
| Bond lengths (Å) | 0.012 | | | | 0.012 |
| Bond angles (°) | 1.296 | | | | 1.438 |

deficiency, which lie on 64 different residues of the polypeptide (Supplementary Fig. 7). Twice as many of the mutation sites are found in the catalytic domain ($n = 38$) as the regulatory domain (20), with the remainder (6) found in the linker region. By proportion, however, the linker region has a higher density (24% of the sequence) of mutation sites than the catalytic (11%) and regulatory (7%) domains. Additionally, a number of sites in the catalytic and regulatory domains are in direct contact with the linker region. Further, the most severe mutations, those found either homozygously or in conjunction with a truncating mutation to result in enzymatic activity below 1.5% of control activity in patient fibroblasts[20], cluster in the catalytic domain and the first two aa of the linker region, most of which are located where the linker meets the catalytic domain (Supplementary Fig. 7). Together, this analysis underscores the importance of the linker region to proper protein function.

**An asymmetric MTHFR dimer with inter-domain flexibility**. The HsMTHFR$_{38-644}$ structure reveals a homodimer (Fig. 4b), consistent with native mass spectrometry (Supplementary Fig. 3b) and previous investigation of mammalian MTHFR by size exclusion chromatography and scanning transmission electron microscopy[9]. It was previously thought that MTHFR homodimerizes in a head-to-tail manner, where the regulatory domain of one subunit interacts with the catalytic domain of the other subunit[13]. Unexpectedly, in our structure dimerization is mediated almost entirely by the regulatory domain (Fig. 4b), although the first ordered residue in chain A (Glu40) is located around 5–6 Å from the regulatory domain of chain B (e.g. Glu553, Arg567). The N-terminal sequence that is either not present (Ser-rich phosphorylation region, aa 1–37) or present but disordered (aa 38–39) in the HsMTHFR$_{38-644}$ structure will likely project

towards the interface of the two regulatory domains (Fig. 4c), and may contribute further to the dimer contacts.

The essential interfacial residues from the regulatory domain are contributed predominantly from the two central β-sheets, including a β-turn (β11–β12), strand β16, and the loop encompassing Asn386–Asn391 (Supplementary Fig. 8), which buries in total ~1330 Å$^2$ of accessible surface. Half of the sites of missense mutations in the regulatory domain causing MTHFR deficiency ($n = 10$, Supplementary Fig. 7) either participate in, or are within two residues of, the dimerization site.

Within the homodimer, each of the two catalytic domains is presented away from the dimeric interface and their active sites are at opposite ends of the overall shape and face away from each other (Fig. 4b). In this arrangement, the catalytic domain is not involved in oligomerization, unlike bacterial and archaeal MTHFR proteins (Supplementary Fig. 9). This said, the N-terminus of the HsMTHFR$_{38-644}$ construct is projecting towards the dimer interface. A direct consequence of the dimeric architecture is that the HsMTHFR catalytic domain displays a large degree of flexibility in relative orientation with the regulatory domain. In fact, this is reflected in our structure whereby the catalytic domain of one dimer subunit (chain A) is ordered, while that of the other dimer subunit (chain B) is highly disordered, to the extent that only main chain atoms of the amino acid 40–58, 129–134 and 155–342 in chain B could be modelled.

**Dynamics of MTHFR observed by solution scattering**. Our HsMTHFR$_{38-644}$ crystal structure has captured the snapshot of an asymmetric dimer whereby the two catalytic domains have different orientations with respect to their own regulatory domains (Supplementary Fig. 10). We applied small-angle X-ray scattering (SAXS) to understand better the different conformational

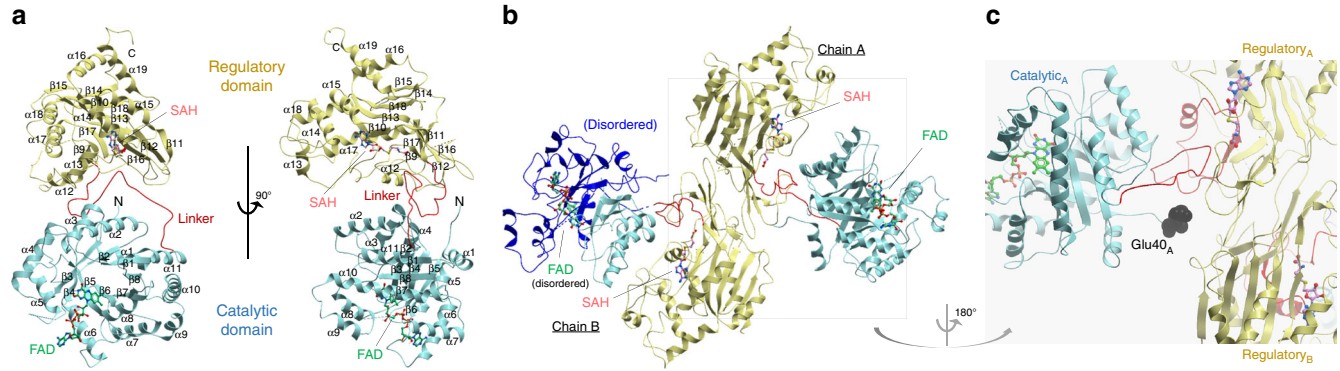

**Fig. 4** Structural overview of *Hs*MTHFR. **a** Orthogonal views of *Hs*MTHFR showing the catalytic domain (cyan), the linker (red) and the regulatory domain (yellow). Bound FAD (green) and SAM (pink) are shown in sticks. Dotted lines indicate disordered regions that are not modelled in the structure. α-Helices and β-sheets are labelled, and correspond to the multiple sequence alignment in Supplementary Figs. 13 and 15 and the topology depicted in Supplementary Fig. 6. **b** Homodimer of *Hs*MTHFR as seen in the crystal. Chain A represents the more ordered protomer. In chain B, the part of the catalytic domain that was poorly ordered is represented in dark blue and only the main chain was modelled. The FAD in this subunit was also partially disordered. **c** Juxtaposition of the N-terminus of *Hs*MTHFR protomer towards both regulatory domains of the homodimer. The first residue observed in the structure, Glu40 of chain A, is shown in black spheres. Other coloured features are as described for panel **a**

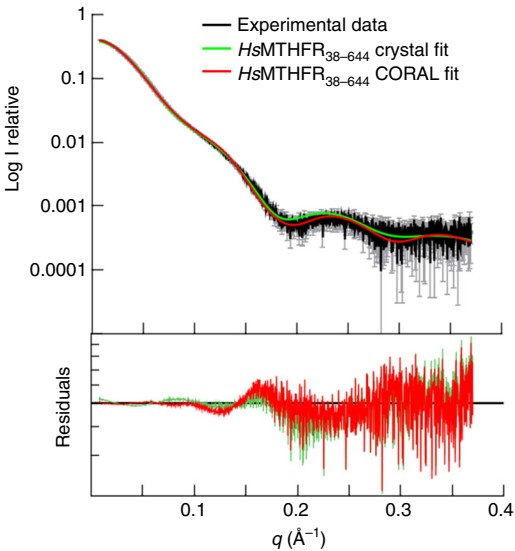

**Fig. 5** SAXS analysis of *Hs*MTHFR$_{38-644}$ and *Hs*MTHFR$_{1-656}$ phosphorylated and dephosphorylated. SAXS analysis of *Hs*MTHFR$_{38-644}$. Experimental scattering profile is shown in black, theoretical scattering curve of the *Hs*MTHFR$_{38-644}$ dimer observed in the crystal is in green and that of the rigid body modelling by CORAL is in red. Chi$^2$ was determined by CRYSOL[53]

variations assumed by the protein in solution. Superimposition of the theoretical scattering curve back-calculated from the crystal structure dimer against experimental data obtained from *Hs*MTHFR$_{38-644}$ in solution revealed a poor fit (Chi$^2$ 14.8; Fig. 5), suggesting this is not the predominant conformation in solution. However, by employing CORAL[21] to simulate relaxation of the relative orientations of the catalytic and regulatory domains (by allowing flexibility in residues 338–345 of the linker), and thus also permitting rigid body movement of these subunits in relative orientation to each other, we obtained a significantly improved fit (Chi$^2$ 5.5; Fig. 5). Thus, consistent with our finding from the crystal structure, *Hs*MTHFR retains a significant degree of intra- and inter-domain conformational flexibility in solution.

To further investigate the influence of phosphorylation on protein conformation, we next collected SAXS data for full-length *Hs*MTHFR$_{1-656}$ as purified (i.e. phosphorylated and bound with

SAM) and treated with CIP (i.e. dephosphorylated and bound with SAH). The experimental scattering curves for as purified and CIP-treated *Hs*MTHFR$_{1-656}$ gave rise to slightly different profiles and derived parameters (Supplementary Fig. 11, Supplementary Table 1), although both protein forms are consistent with a dimeric configuration. These data were further corroborated by charge radius analysis of native phosphorylated and dephosphorylated *Hs*MTHFR$_{1-656}$ by electrospray ionization mass spectrometry, showing that the charge-distribution of protein ions is shifted between the two protein forms. This may indicate a conformational change equivalent to a 0.5% change in radius (Supplementary Fig. 12). Together, we interpret these results to suggest that the phosphorylated SAM-bound form of the protein may present a different conformation to the dephosphorylated SAH-bound form, which merits future investigation using alternative methods.

**Subtle features provide for eukaryotic NADPH specificity.** The MTHFR catalytic domain adopts a TIM-barrel structure evolutionarily conserved across all kingdoms. In addition to *Hs*MTHFR$_{38-644}$, we further determined the catalytic domain structure of the yeast homologue MET12 (*Sc*MET12$_{1-301}$) to 1.56 Å resolution (Table 2). This enables a structural comparison across mammalian (*Hs*MTHFR), low eukaryotic (*Sc*MET12) and bacterial (*E. coli*, *H. influenzae*, *T. thermophilus*) orthologues. Consistent with their sequence conservation (Supplementary Fig. 13), the catalytic domains have highly superimposable folds (main chain RMSD: 1.85 Å), although distinct local differences are found in low homology loop regions (Fig. 6a, 1–2) and helices (Fig. 6a, 3–4). Additionally, the first helix of the catalytic domain (α1) is observed in different orientations among these structures (Supplementary Fig. 14). There is sequence divergence of helix α1 among prokaryotes, lower and higher eukaryotes (Supplementary Fig. 15). In *Hs*MTHFR (which contains, in its biological sequence but not present in the crystallized construct, the serine-rich phosphorylation region N-terminal to the catalytic domain), *Sc*MET12 and *T. thermophilus* MTHFR, this helix α1 is projected towards the interface between catalytic and regulator domains.

In *Hs*MTHFR$_{38-644}$, clear electron density for FAD was observed in the TIM barrel of chain A (Supplementary Fig. 5a). However, there is high disorder in the TIM-barrel of chain B particularly around the FAD binding site, implying a low ligand occupancy of the ligand, although native mass spectrometry of

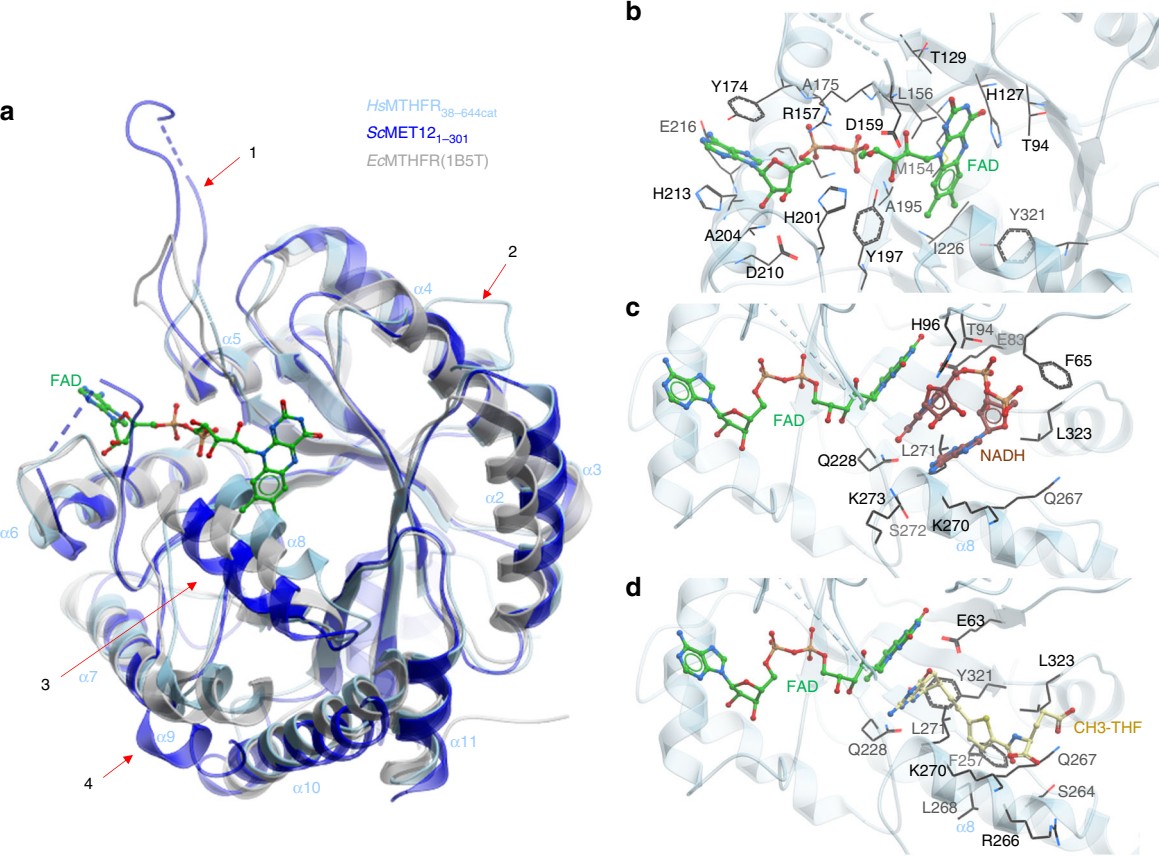

**Fig. 6** Structural examination of the $HsMTHFR_{38-644}$ catalytic domain. **a** Structural alignment of $HsMTHFR_{38-644}$ (cyan) with $EcMTHFR$ (grey) and $ScMET12_{1-301}$. Four sites of important differences are indicated by arrows (1–4). α-helices of $HsMTHFR_{38-644}$ are indicated for orientation. **b** Binding pocket of FAD. FAD is shown in green sticks, residues contributing to FAD binding are labelled and shown in black sticks. **c** Binding pocket of NAD(P)H. NADH is taken from an overlay of $EcMTHFR$ (PDB: IZRQ) with $HsMTHFR_{38-644}$ but for clarity $EcMTHFR$ is not shown. FAD is shown in green sticks, NADH in brown sticks, residues expected to contribute to NADH binding are labelled and shown in black sticks. **d** Binding pocket of $CH_3$–THF. $CH_3$–THF is taken from an overlay of $EcMTHFR$ (PDB: 2FMN) with $HsMTHFR_{38-644}$ but for clarity $EcMTHFR$ is not shown. FAD is shown in green sticks, $CH_3$–THF in yellow sticks, residues expected to contribute to $CH_3$–THF binding are labelled and shown in black sticks

the crystallized construct indicated two FADs bound per homodimer (Supplementary Fig. 3b). Analysis of the FAD binding residues in $HsMTHFR$ chain A (Fig. 6b) reveals perfect overlap with those predicted from the $EcMTHFR$ structure[4]. These include Thr129, Arg157, Ala175 and Ala195 of $HsMTHFR$, which were associated with in vitro FAD responsiveness when mutated in severe MTHFR deficiency[20,22] (Fig. 6b; Supplementary Fig. 7).

The bi bi kinetic mechanism of MTHFR necessitates the electron donor NAD(P)H and substrate $CH_2$-THF to interact in turn with FAD for transfer of the reducing equivalents, and hence to share the same binding site. In our structures, the FAD ligand adopts a conformation poised to expose the *si* face of the isoalloxazine ring for the incoming NAPDH and $CH_2$-THF. However, instead of trapping the electron donor or substrate (despite multiple attempts at co-crystallization), the binding site in $ScMET12_{1-301}$ and subunit A of $HsMTHFR_{38-644}$ is blocked by a crystal packing interaction from a nearby symmetry mate, making π–π stacking interactions with the FAD ligand (Supplementary Fig. 16). By contrast, no crystal packing interaction is found in the chain B binding site of $HsMTHFR_{38-644}$, explaining the overall mobility and disorder of its catalytic domain.

Superimposing the $HsMTHFR_{38-644}$ structure with that of $EcMTHFR$ bound with NADH (Fig. 6c) and $CH_3$-THF (Fig. 6d) demonstrates that the human enzyme has largely preserved the

same shared binding site found in prokaryotes, with Gln228, Gln267, Lys270, Leu271 and Leu323 likely to be important for interacting with both NAD(P)H and $CH_3$-THF. $EcMTHFR$ preferentially utilizes NADH[23], and its NADH-bound structure reveals a highly uncommon bent conformation[24] for the electron donor, where the nicotinamide ring stacked over the adenine base mediates π–π interactions[7]. Our activity assay of $HsMTHFR_{38-644}$ and $HsMTHFR_{1-656}$ clearly demonstrates an ~100-fold preference for NADPH compared to NADH as an electron donor (Table 1), in agreement with previous enzyme studies from pig[11,25] and rat[11] MTHFRs.

Within the $HsMTHFR$ active site, we did not identify any obvious differentiating features surrounding the modelled NADH, which could indicate how the extra 2′-monophosphate group on the NADPH ribose is accommodated (Supplementary Fig. 17). It is also unclear if $HsMTHFR$ actually binds NADPH in a similar manner as NADH for $EcMTHFR$, considering there is only one report in the literature documenting a compact stacked conformation for NADPH[26]. Modelling an NADPH ligand with such a stacked conformation onto the $HsMTHFR_{38-644}$ structure reveals severe steric clashes with helix α8 (Supplementary Fig. 17), which creates the floor of the NAD(P)H binding site (e.g. via Gln267, Lys270 and Leu271). Helix α8 is poorly aligned with bacterial and low eukaryotic orthologues in both amino acid sequence (Supplementary Fig. 13) and structural topology

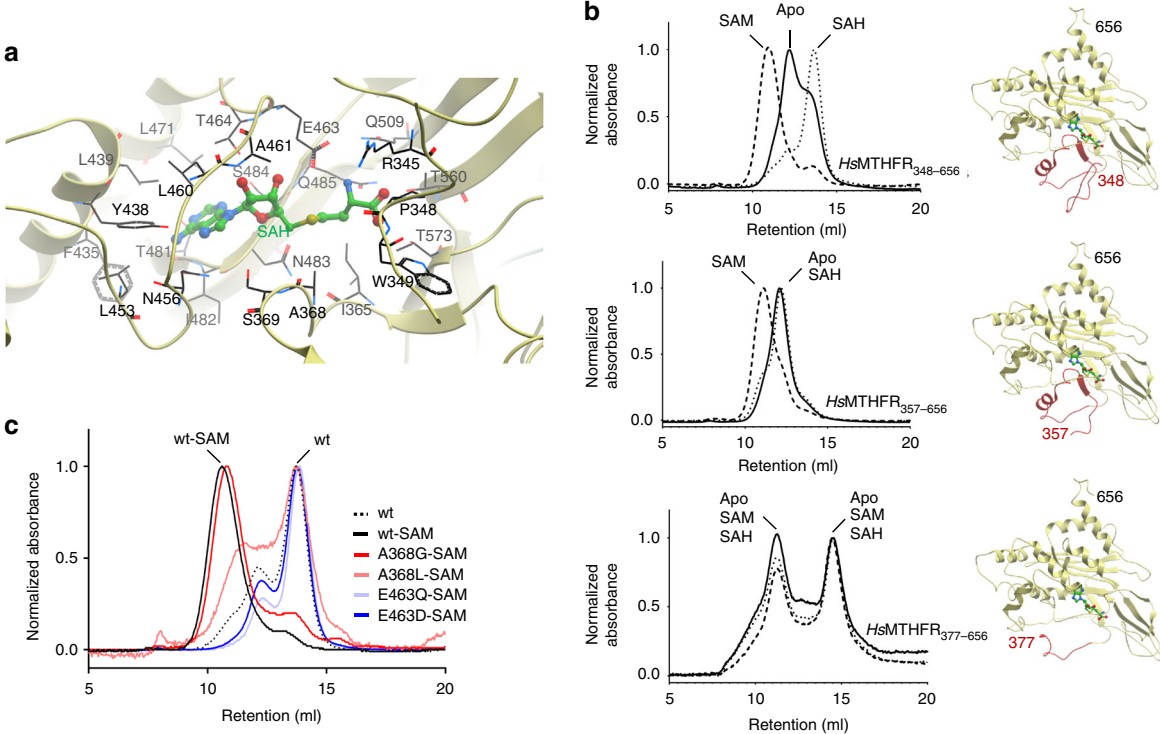

**Fig. 7** SAH/SAM binding and conformational change. **a** The SAH binding site. Amino acids that contribute to binding are labelled and shown in black sticks. SAH is shown in green sticks. **b** Size exclusion chromatography of *Hs*MTHFR with various N-terminal truncations following incubation with SAM (dashed lines), SAH (dotted lines), or buffer (apo; solid line). For each N-terminally truncated construct, the corresponding structure is shown. **c** Size exclusion chromatography of *Hs*MTHFR₃₄₈₋₆₅₆ proteins without (wt) or with (wt-SAM) pre-incubation with SAM. Mutated *Hs*MTHFR₃₄₈₋₆₅₆ proteins were pre-incubated with SAM

(Fig. 6a). The equivalent helix in *Ec*MTHFR harbours the residue Phe223, which is crucial to NADH binding[7] and moves to accommodate substrate release[5]. Notably, this residue is not conserved in *Hs*MTHFR and *Sc*MET12, replaced by Gln267 and Ala229, respectively. (Supplementary Fig. 13). Therefore, given its position and mobility, we propose that residue(s) on helix α8 in *Hs*MTHFR may play a role in the specificity for NADPH and likely also substrate binding/release.

**A novel fold for the SAM-binding regulatory domain**. The *Hs*MTHFR₃₈₋₆₄₄ structure provides a view of the 3D arrangement of the regulatory domain unique to eukaryotic MTHFR. The core of this fold comprises two mixed β-sheets of five strands each (β9↑-β17↑-β16↓-β12↑-β11↓ and β10↓-β13↑-β18↓-β14↑-β15↓) (Supplementary Fig. 6). Strand β10 from one sheet forms a continuous segment with β11 from the other sheet, and similarly β12 from one β-sheet continues onto β13 of the other sheet. The threading of the two central β-sheets are interspersed with three loop extensions containing different numbers of α-helices (α12–α15, α16, and α17–α18). To the best of our knowledge, the MTHFR regulatory domain represents a unique SAM binding architecture distinct from the 18 known classes of SAM-dependent methyltransferases and non-methyltransferases[27] (Supplementary Fig. 18). Further, a DALI search of this domain[28] did not yield any structural homologue, and we found no existing annotation in PFAM/CATH/SCOP databases and no sequence for this domain beyond eukaryotic MTHFR homologues. Therefore, this appears to be a novel fold utilized only by MTHFR for SAM binding/inhibition.

In our structure, SAH is bound in an extended conformation within the part of the regulatory domain (Fig. 7a) that faces the catalytic domain. Indeed, part of the binding site is constituted by the linker region itself. The ligand is sandwiched between the loop segment preceding α15 (N₄₅₆DEPLAAET₄₆₄) and the first strand β10 (T₄₈₁INSQ₄₈₅) of the central β-sheets, where a number of conserved residues are found. For example, Thr481 (conserved in 96% of 150 orthologues; Consurf[29]) and Ser484 (98%) hydrogen-bond to the SAH adenine moiety, while Glu463 (99%) and Thr464 (62%) fixate the ribose hydroxyl groups. The strongest sequence conservation in the SAH binding site is found around the homocysteine moiety, including Pro348 (invariant) and Trp349 (99%) from the linker region, as well as Thr560 and Thr573 (both invariant) at the start and end of the β15-β16 turn. The SAH homocysteine sulphur atom is loosely contacted by Glu463 (3.8 Å) and Ala368 (3.7 Å). SAM is expected to bind to the same site in the regulatory domain, in a similar extended configuration as SAH and requiring the same set of binding residues. However, the additional methyl group in the sulphonium centre of SAM would create a steric clash to the Ala368 position of the structure (inter-residue distance ~2.0 Å between heteroatoms, and <1.5 Å between hydrogen atoms) (Supplementary Fig. 19). Although not strictly conserved (45% of 150 orthologues), conservation of Ala368 follows a similar evolutionary pattern as the MTHFR domain organization (Fig. 1): in higher animals alanine is invariant; lower animals may accommodate a serine; while lower eukaryotes often incorporate a bulky residue (e.g. lysine) (see Supplementary Fig. 15). Therefore, in higher organisms such as humans, SAM binding likely results in conformational rearrangement of the loop region containing Ala368 to accommodate its methyl moiety.

**The linker mediates SAM-dependent conformational change.** Since there is no direct interface between the active site of the catalytic domain and the regulatory domain (Fig. 4), SAM binding must elicit enzymatic inhibition via a conformational change propagated from the regulatory to catalytic domain. The most likely effector of this conformational change is the extended linker region (defined as aa 338–362), since it makes multiple contacts to both the regulatory and catalytic domains (Fig. 4) and forms part of the SAM/SAH binding site (Fig. 7a). To investigate the potential of this region to elicit conformational change following SAM binding, we generated recombinant $Hs$MTHFR proteins consisting of the regulatory domain alone attached to progressively shorter linker regions, where the N-terminus of these constructs would become Pro348 ($Hs$MTHFR$_{348-656}$), Arg357 ($Hs$MTHFR$_{357-656}$) and Arg377 ($Hs$MTHFR$_{377-656}$) (Fig. 7b; Supplementary Fig. 2). All three constructs are sufficient to bind SAM and SAH, as demonstrated by dose-dependent increases in thermostability by differential scanning fluorimetry when exposed to increasing concentrations of each ligand (Supplementary Fig. 20a). This again reinforces the catalytic and regulatory domains as separate binding modules for their cognate ligands (FAD/NADPH/CH$_3$-THF vs SAM/SAH respectively).

We employed analytical size exclusion chromatography (aSEC) as a means to study solution behaviour of the MTHFR regulatory domain in response to SAM/SAH binding. Exposure of MTHFR$_{348-656}$ to either SAH or SAM resulted in shifts of elution volume ($V_e$) compared to as purified (apo-) protein (Fig. 7b), in contrast to $Hs$MTHFR$_{1-656}$ and $Hs$MTHFR$_{38-644}$, which did not show changes in $V_e$ despite SAH or SAM binding (Supplementary Fig. 21). Importantly, for MTHFR$_{348-656}$, SAM resulted in a leftward $V_e$ shift (suggestive of a larger hydrodynamic volume) and SAH a rightward shift (suggestive of a smaller hydrodynamic volume) (Fig. 7b). By contrast, MTHFR$_{357-656}$ showed a shift in $V_e$ only when exposed to SAM, and MTHFR$_{377-656}$ did not change when exposed to either ligand (Fig. 7b). A similar pattern of results were observed when using purified recombinant mouse MTHFR of the same protein boundaries (Supplementary Fig. 20a and b). Therefore, we conclude that residues within 357–377 must contribute to change of protein state upon SAM binding, which we interpret as a change in conformation.

Next we carried out site-directed mutagenesis to define residues involved in SAM binding, and/or SAM-mediated conformational change as observed in the aSEC experiment. We reasoned that mutation of Glu463 (which hydrogen-bonds a ribose oxygen) could lead to loss of SAH/SAM binding, and thus conformational change. Indeed, conservative mutation of Glu463 to either aspartate (p. E463D) or glutamine (p.E463Q) on MTHFR$_{348-656}$ resulted in protein that could no longer bind SAM (Supplementary Fig. 20c), nor change conformation in its presence (Fig. 7c). We further hypothesized that mutation of Ala368 (in close proximity to the SAM/SAH sulphonium centre) to a smaller residue (glycine: p. A368G) may not have an effect on binding or conformational change, while mutation to a larger residue (leucine: p.A368L) might reduce the ability of the linker region to sense SAM binding. Correspondingly, p.A368L resulted in protein which retained the ability to bind SAM, but was less sensitive to change in its presence, while p.A368G did not change either of these properties. (Fig. 7c, Supplementary Fig. 20c). These experiments conclusively pinpoint Glu463 as crucial to SAM binding and Ala368 to SAM sensing, representing a mechanism that could transmit a ligand-bound signal from regulatory to catalytic domain of the protein.

## Discussion

Catalytic regulation by phosphorylation and SAM binding distinguishes human MTHFR from its bacterial (which do not have a phosphorylation or SAM binding regions) and lower eukaryotic (which do not have a phosphorylation region) counterparts. Until now, the molecular basis of how these two allosteric events modulate the catalytic machinery was entirely unknown, due to the absence of a structural context. Now, our structure-guided study has provided two major discoveries in this area: (1) identification of an extensive linker region that functionally connects SAM-binding in the regulatory domain with inhibition in the catalytic domain and (2) demonstration of the concerted effects of phosphorylation and SAM binding, individually mediated by regions more than 300 amino acids apart (Supplementary Fig. 22).

We mapped the entire phosphorylation landscape of $Hs$MTHFR, revealing phosphorylated Ser/Thr not only at the far N-terminus ($n = 11$) as predicted from the sequence but also within the catalytic (3) and regulatory (2) domains. Many of the N-terminal phosphorylation sites identified are consistent with previous mutation analysis[14], including Thr34 (refs. [14–16]). The phosphorylated residues detected in the catalytic and regulatory domains were not reported before. Interestingly, two phosphorylated serines are located within the FAD binding site, although their physiological significance is currently unclear. Contrary to the recent observation of Li et al.[17] we did not identify phosphorylation of Thr549.

An important finding with regard to MTHFR phosphorylation is that it does not directly alter the catalytic parameters of the enzyme, as determined by a sensitive HPLC-based activity assay. Perhaps this is not too surprising, since the first ordered residue of the structure, Glu40 (i.e. immediately following the phosphorylation region aa 1–37), is far removed from the catalytic site. Instead, MTHFR phosphorylation exerts a long-range influence on the SAM binding status at the regulatory domain some 300 amino acids away, by causing an increased sensitivity of the enzyme to SAM inhibition, but with no overall changes on total SAM inhibition. Phosphorylation likely enhances SAM sensitivity in two interdependent ways. Firstly, it enables protection of bound SAM from spontaneous degradation to SAH, a phenomenon widely observed for SAM-bound enzymes in vitro[30] and in crystallo[31], to avoid dis-inhibition by SAH. Secondly, phosphorylation could induce a conformational change to the protein that primes an inhibition ready state. The SAM $Ki$ differences between phosphorylated and dephosphorylated protein, while relatively small (2–3 vs. 6–7 µM), are likely to be physiologically relevant. Intracellular SAM concentrations are reported to be 1–3 µM in human cells[32,33], and the mTORC1 linked starvation sensor SAMTOR, which recognizes SAM for nutrient sensing, has a SAM dissociation constant of 7 µM[34].

It is not immediately clear if global phosphorylation or phosphorylation of only specific residues contributed to the results we found. While 16 phosphorylation sites of $Hs$MTHFR are identified in this study, intact mass analysis shows that only 9–10 phosphorylations were present at a time, implying a high degree of heterogeneity. While multiply phosphorylated peptides were observed, they were not quantified using our methods. Identification of MTHFR proteoforms to assess whether phosphosites occur in combinations, perhaps using top-down MS[35] would be very interesting, and may constitute the basis of future studies. Evolutionary conservation of the identified phosphorylation sites varies from absolute invariance through to yeast (e.g. Ser394), to poor conservation even among animals (e.g. Ser9 and Ser10) (Supplementary Fig. 15). Truncated recombinant $Hs$MTHFR$_{38-644}$ was not identified to be phosphorylated by mass spectrometry or crystallography, suggesting that phosphorylation at the far N-terminus primes the other phosphorylation events within the catalytic and regulatory domains. This is consistent with previous observations[14–16] that removal of Thr34 results in

non-phosphorylated protein in vitro. It remains to be seen whether phosphor-Thr34 alone, or other sites at the Ser-rich region, primes other phosphorylation events in vivo.

We identified the MTHFR regulatory domain to constitute a novel SAM binding fold whose appendage to the well-conserved catalytic TIM-barrel is a relatively recent and contained evolutionary event. A similar phenomenon of domain organization is found in several eukaryotic enzymes, for example those involved in amino acid metabolism (e.g. cystathionine β-synthase, CBS[36]; phenylalanine hydroxylase, PAH[37]), whereby the additional metabolite-binding modules, not found in their bacterial counterparts, serve to fine-tune catalysis in response to the more intricate higher eukaryotic metabolic and signalling cues. We propose that MTHFR belongs to this class of allosteric enzymes that share a common mechanism—to regulate catalysis through steric sequestration of the catalytic site, in a ligand-dependent manner (SAM for MTHFR and CBS; phenylalanine for PAH).

Although our crystal structure represents a static snapshot of the enzyme state (likely a dis-inhibited state due to SAH binding), the potential for inter-domain conformational changes is suggested by the following data. Firstly, the two chains in the crystal asymmetric unit show varying intrinsic order of the catalytic domain with respect to the regulatory domain. Secondly, SAXS and native mass spectrometry analysis between SAM-bound phosphorylated protein and SAH-bound dephosphorylated protein hint at subtle, but distinguishable changes to the protein dimensions. It remains to be determined from future studies whether MTHFR undergoes ligand-dependent conformational changes to rearrange domain orientation for catalysis, like PAH and CBS. If this should be the case, the MTHFR linker region will likely contribute to this role, as it makes extensive contacts with both the regulatory domain (e.g. SAM binding site) and catalytic domain (e.g. helices α3 and α4). This is also consistent with the concentration of deleterious disease mutations found in this region. Additional genetic data from our lab are in accord[20], as patient fibroblasts homozygous for p.His354Tyr, a linker residue which contacts helix α3 in catalytic domain, exhibited a five-fold decrease in $K$i for SAM.

So in what aspects could the SAM-bound signal influence the catalytic domain, seeing that its kinetic parameters remain largely unaltered? One possibility is an effect on the stability or integrity of FAD, the essential cofactor. We observed that supplementation with FAD enabled rescue of activity to our recombinant MTHFR. This is indicative of cofactor loss, in agreement with previous findings[13], and suggestive of FAD being only loosely bound, as exemplified in chain B of our structure. Furthermore, a number of MTHFR mutations[20,22] and polymorphisms[13] are shown to affect FAD responsiveness. It is therefore possible that the inter-domain flexibility we observed, communicated by the SAM-bound signal, would alter the orientation of catalytic domains with respect to the rest of the protein, in a similar manner as the multi-domain enzymes CBS and PAH[38]. Such structural conformations are supported by overlays between chains A and B in our structure, and between apo- and holo-subunits in *Tt*MTHFR[8]. In the case of MTHFR, the active site could be more sequestered (leading to FAD bound) or more exposed and mobile (leading to FAD loss) as a consequence.

The SAM/SAH ratio is regarded as an indicator of a cell's methylation potential and is a crucial indicator of the cells' capacity to perform DNA methylation or create compounds which require methyl groups for assimilation. In the face of a low SAM/SAH ratio, meaning methyl donor deficiency, MTHFR is dis-inhibited, increasing the production of CH₃-THF to improve throughput of the methionine cycle and replenish SAM levels. Conversely, a high SAM/SAH ratio means abundant methylation capacity, in which case SAM-mediated allosteric inhibition of MTHFR turns off CH₃-THF production, thereby lowering methionine cycle activity and concomitantly generation of SAM. This on/off switch is especially powerful at high SAM levels, as illustrated by almost complete inhibition of recombinant *Hs*MTHFR at >200 μM SAM. Although these types of concentrations are unlikely to be seen inside the cell, *Hs*MTHFR has been further outfitted with a dimmer switch, whereby protein phosphorylation increases sensitivity to SAM-mediated inhibition at normal (1–10 μM) cellular SAM levels. In this regard, phosphorylation allows linkage of the methionine cycle to other cellular pathways (e.g. cell cycle) through specific kinase activities (as suggested by Zhu et al.[16] and Li et al.[17]).

The clear correlation we observed between phosphorylated MTHFR with SAM binding in solution (vs dephosphorylated MTHFR and SAH binding) leads us to interpret that the two regulatory properties act in concert. In fact, the architecture of the *Hs*MTHFR homodimer is smartly tailor-made to facilitate this correlation. (1) The dimeric interface is entirely constituted by the regulatory domain to form a scaffold, while leaving each SAM binding site on a different face for its sensing and signal transmission functions. (2) The lack of contacts between intermonomeric catalytic domains allows for the intrinsic intramonomeric mobility with respect to the regulatory domains for signal propagation. (3) Importantly, this dimeric configuration brings the N- and C-termini of the polypeptide in proximity, projecting the phosphorylation region close to the regulatory domain dimer interface.

In summary, we provide a structural view of a eukaryotic MTHFR, pointing to the linker region playing a direct role in allosteric inhibition following SAM binding, and phosphorylation as a means to modulate SAM inhibition sensitivity. Modulating such finite control towards the level of a key metabolite may be of pharmacological interest, including in cancer metabolism[39,40]. Our work here constitutes a strong starting point for future, more precise investigation by structural, biochemical and cellular studies, for example towards: identification of the kinase(s) responsible for MTHFR phosphorylation in vivo; combining different structural methods to delineate conformational changes of the entire protein, and revealing the molecular basis of its specificity over NADPH.

## Methods

**Recombinant production of MTHFR.** Numbering of the nucleotide changes follows the nomenclature of NM_005957.4, which places the A of the ATG initiation codon as the +1 nucleotide. The protein is numbered according to NP_005948.3. For *E. coli* (BL21(DE3)R3) expression, DNA fragments encoding human MTHFR (IMAGE: 6374885), mouse MTHFR (IMAGE: 6834886) and yeast MET12 (clone: ScCD00096551 from Harvard Medical School) harbouring different N- and C-terminal boundaries were amplified and subcloned in pNIC28-Bsa4 vector (accession number: EF198106) in-frame with a tobacco etch virus protease cleavable N-terminal His₆-tag. For baculovirus expression, DNA fragments encoding human MTHFR (IMAGE: 6374885) harbouring different N- and C-terminal boundaries were cloned into the pFB-CT10HF-LIC vector (Addgene plasmid: 39191) in-frame with a tobacco etch virus protease cleavable C-terminal flag/His₁₀-tag. Site-directed mutations were constructed using the QuikChange mutagenesis kit (Stratagene) and confirmed by sequencing. All primers are available upon request. Proteins expressed in *E. coli* were purified by affinity (Ni-Sepharose; GE Healthcare) and size exclusion (Superdex 200; GE Healthcare) chromatography. Proteins expressed in insect cells in Sf9 media (ThermoFisher) were purified by affinity (Ni-NTA, Qiagen) and size exclusion (Superdex 200) chromatography, followed by cleavage of the C-terminal tag by His-tagged tobacco etch virus protease (1:20 mass ratio) overnight at 4 °C and re-passaged over Ni-NTA resin. Selenomethionine (SeMet)-derivatized proteins were expressed using Seleno-Methionine Medium Complete (Molecular Dimensions) and purified as above.

**Crystallization and structure determination.** Purified native *Sc*MET12₁₋₃₀₂ as well as SeMet-derivatized and native *Hs*MTHFR₃₈₋₆₄₄ were concentrated to 15–20 mg mL⁻¹, and crystals were grown by sitting drop vapour diffusion at 20 °C. The mother liquor conditions are summarized in Table 2. Crystals were cryo-protected in mother liquor containing ethylene glycol (25% v/v) and flash-cooled in liquid nitrogen. X-ray diffraction data were collected at the Diamond Light Source and

processed using XIA2 (ref. [41]). The $Hs$MTHFR$_{38-644}$ structure was solved by selenium multi-wavelength anomalous diffraction phasing using autoSHARP[42], and subjected to automated building with BUCCANEER[43]. The SeMet model was used to solve the native structure of $Hs$MTHFR$_{38-644}$ by molecular replacement using PHASER[44]. This structure was refined using PHENIX[45], followed by manual rebuilding in COOT[46]. Phases for $Sc$MET12$_{1-302}$ were calculated by molecular replacement using 3APY as model. Atomic coordinates and structure factors for both $Sc$MET12$_{1-302}$ (accession code: 6FNU) and $Hs$MTHFR$_{38-644}$ (accession code: 6CFX) have been deposited in the Protein Data Bank. Data collection and refinement statistics are summarized in Table 2. The final models of $Hs$MTHFR$_{38-644}$ and $Sc$MET12$_{1-302}$ contain respectively 95.6% and 98.6% in the favoured region, 4.3% and 1.1% in the allowed region, and 0.17% and 0.36% in the outlier region of the Ramachandran plot.

**MTHFR assay.** All enzymatic assays, including SAM inhibition and thermolability, were performed using the physiological forward assay described by Suormala et al.[18] with modifications as described by Rummel et al.[47] and Burda et al.[20,22]. Only minor adaptations were made for use with pure protein, including reducing the assay time to 7 min and the addition of bovine serum albumin (BSA) to keep purified proteins stable. Prior to assay, purified proteins were diluted from 15–20 to 1 mg mL$^{-1}$ in 10 mM HEPES-buffer pH 7.4, 5% glycerol and 500 mM NaCl followed by successive dilutions of 1:100 and 1:32 in 10 mM potassium phosphate, pH 6.6 plus 5 mg mL$^{-1}$ BSA, to a final MTHFR concentration of 312.5 ng mL$^{-1}$. Specific activity was measured in 50 mM potassium phosphate buffer, pH 6.6, under saturating substrate concentrations (75 µM methyleneTHF, Eprova AG; 200 µM NADPH; Sigma-Aldrich) with and without the addition of FAD (75 µM; Sigma-Aldrich). The $K_M$ for NAPDH was determined by varying its concentration between 10 and 250 µM in the presence of 75 µM CH$_2$-THF. The $K_M$ for CH$_2$-THF was determined by varying its concentration between 2.5 and 100 µM in the presence of 75 µM NAPDH. All $K_M$ values were derived using a non-linear fit of Michaelis–Menten kinetics by GraphPad Prism (v6.07). For SAM inhibition, purified SAM[48] was used. The $K_i$ was estimated following a plot of log(inhibitor) vs. response and a four parameter curve fit as performed by GraphPad Prism (v6.07).

**In-solution analysis.** Analytical gel filtration was performed on a Superdex 200 HiLoad 10/30 column (GE Healthcare) pre-equilibrated with 10 mM HEPES pH 7.5, 150 mM NaCl and 5% glycerol in the presence or absence of 250 µM SAH or SAM (both Sigma-Aldrich). The column was calibrated using carbonic anhydrase (29 kDa), BSA (66 kDa), alcohol dehydrogenase (150 kDa) and apoferritin (443 kDa) as protein standards. Differential scanning fluorimetry was used to assay shifts in melting temperature caused by ligand binding in a 96-well PCR plate using an LC480 light cyler (Roche). Each well (20 µL) consisted of protein (0.1 mg mL$^{-1}$), SYPRO-Orange (Invitrogen) diluted 1000× and buffer (10 mM HEPES pH 7.5, 500 mM NaCl) in the presence of 0–250 µM SAM or SAH. Fluorescent intensities were measured from 20 to 78 °C with a ramp rate of 1 °C min$^{-1}$. The temperature shifts, $\Delta T_m^{obs}$, were calculated using a Boltzmann sigmoidal fit[49], and half-maximal effective ligand concentrations, AC$_{50}$, for each ligand was fit to a one-site total binding model, both using GraphPad Prism (v6.07).

**Small-angle X-ray scattering.** SAXS experiments for the $Hs$MTHFR$_{38-644}$ and $Hs$MTHFR$_{1-656}$ (phosphorylated and dephosphorylated) were performed at 0.99 Å wavelength Diamond Light Source at beamline B21 coupled to the Shodex KW403-4F size exclusion column (Harwell, UK) and equipped with Pilatus 2 M two-dimensional detector at 4.014 m distance from the sample, $0.005 < q < 0.4$ Å$^{-1}$ ($q = 4\pi \sin\theta \lambda^{-1}$, $2\theta$ is the scattering angle). The samples (20 mg mL$^{-1}$) were in a buffer containing 20 mM Hepes-NaOH pH 7.5, 0.5 mM TCEP, 150 mM NaCl, 2% glycerol and 1% sucrose. SAXS measurements were performed at 20 °C, using an exposure time of 3 s frame$^{-1}$. To perform dephosphorylation, as purified (phosphorylated) $Hs$MTHFR$_{1-656}$ was added with 10 mM Mg-acetate in the above buffer and incubated with 100U calf alkaline phosphatase at 37 °C for 2 h. SAXS data were processed and analysed using the ATSAS program package[21] and Scatter (http://www.bioisis.net/scatter). The radius of gyration Rg and forward scattering $I(0)$ were calculated by Guinier approximation. The maximum particle dimension $D_{max}$ and $P(r)$ function were evaluated using the programme GNOM[50]. To demonstrate the absence of concentration-dependent aggregation and interparticle interference in the both SAXS experiments, we inspected Rg over the elution peaks and performed our analysis only on a selection of frames in which Rg was most stable. Overall, such stability of Rg over the range of concentrations observed in the SEC elution indicates that there were no concentration-dependent effects or interparticle interference. Ten runs of CORAL rigid body modelling were performed by defining residues 338–345 as a flexible linker and allowing the catalytic subunits to move while keeping the regulatory subunits fixed.

**Denaturing intact mass analysis.** Reversed-phase chromatography was performed in-line prior to mass spectrometry using an Agilent 1290 uHPLC system. Concentrated protein samples were diluted to 0.02 mg mL$^{-1}$ in 0.1% formic acid and 50 µL was injected on to a 2.1 mm × 12.5 mm Agilent Zorbax 5um 300SB-C3 guard column housed in a column oven set at 40 °C. The solvent system used consisted of 0.1% formic acid in LC-MS grade water (solvent A) and 0.1% formic acid in LC-MS grade methanol (solvent B). Chromatography was performed as

follows: initial conditions were 90% A and 10% B and a flow rate of 1.0 mL min$^{-1}$. A linear gradient from 10% B to 80% B was applied over 35 s. Elution then proceeded isocratically at 95% B for 40 s followed by equilibration at initial conditions for a further 15 s. Protein intact mass was determined using a 6530 QTOF mass spectrometer (Agilent). The instrument was configured with the standard ESI source and operated in positive ion mode. The ion source was operated with the capillary voltage at 4000 V, nebulizer pressure at 60 psig, drying gas at 350 °C and drying gas flow rate at 12 L min$^{-1}$. The instrument ion optic voltages were as follows: fragmentor 250 V, skimmer 60 V and octopole RF 250 V.

**Native mass spectrometry.** Mass spectrometry of MTHFR under native conditions was performed on the same MS instrument, and is described in detail elsewhere[51]. Briefly, 50 µg of protein was desalted and exchanged into 50 mM ammonium acetate pH 6.5 using three rounds of size exclusion spin column purification ("Micro BioSpin 6", Biorad) following the manufacturer's instructions. Approximately 50 µL was infused directly into the mass spectrometer via syringe pump at a rate of 6 µL min$^{-1}$. The instrument was fitted with a standard source using a nebulizer pressure of 17 psi, drying gas flow rate 5 L min$^{-1}$ and drying gas temperature 325 °C. The instrument was operated in positive ion, 1 GHz detector mode with fragmentor voltage 430 V, nebulizer pressure set to 17 psi, and collision gas pressure and energy at 0. Spectra were deconvoluted using the MaxEnt functionality within the Masshunter software package (Agilent) and multimeric states were determined manually using a charge table. The radius for native protein charge states was determined using the calculation $z = R^{1.5}$ or $R = z \times 10^{1/1.5}$ where $z$ is the charge state and $R$ is the radius in Ångstroms if the protein is assumed to be a sphere[52]. To allow comparison between spectra, ion intensities were normalized by dividing each value by the largest in that spectrum. Normalized ion intensity was plotted against charge radius for each ion.

**Phosphorylation mapping.** Between 20 and 100 µg MTHFR was reduced in 100 µL of 100 mM ammonium bicarbonate buffer, pH 7.5 by addition of 1 µL of 1 M DTT and incubation at 56 °C for 40 min. Alkylation was performed by addition of 4 µL of saturated iodoacetamide solution and incubated at room temperature in the dark for 20 min. Endoprotease digestion was performed using either trypsin, Smart Digest trypsin (Thermo) or pepsin. Trypsin in 100 mM ammonium bicarbonate buffer, pH 7.5 was added in the ratio 20:1 w/w and incubated at 37 °C overnight. Smart Digest trypsin was incubated at 70 °C for 1 h following the manufacturer's instructions. Pepsin digestion was performed in 0.1 M HCl in the ratio 20:1 w/w and incubated at 37 °C overnight. LC-MSMS analyses were performed using both whole endoprotease digests and metal oxide-affinity enriched samples. Metal oxide-affinity enrichment for phosphopeptides was performed using home made spin columns containing a mixed bed of 2.5 mg Titansphere TiO$_2$ chromatography matrix (GL Sciences) and 2.5 mg ZrO$_2$ powder. Non-phospopeptides were eluted using 80% ACN, 300 mg mL$^{-1}$ DHB, 0.1% TFA. Phosphopeptides were eluted in 25% ammonium hydroxide, 40% ACN. Samples were dried-down by rotary evaporation and re-suspended in 5 µL of 2% ACN, 0.1% FA prior to LC-MSMS.

Analysis was performed using a Dionex U3000 nanoHPLC coupled to a Bruker Esquire HCT ion trap mass spectrometer. Peptides were separated using a 200 µm × 5 cm Pepswift PS-DVB monolithic column (Thermo, USA). A gradient was developed from 2 to 17% B over 4 min, then 17–42% B over 2 min. The column was washed at 92% B for 1 min and finally equilibrated at 2% B for 6 min at a flow rate of 2.5 µL min$^{-1}$. Buffer A was 0.1% formic acid in LC-MS grade water: Buffer B was 0.1% formic acid, 80% LC-MS grade ACN. MSMS was performed in data-dependent mode with a scan rate of 26,000 m z$^{-1}$ s$^{-1}$ with three precursors per MS1 scan and active exclusion for 20 s. Charge state selection was +1, +2 and +3. Automated data analysis was performed using Data Analysis v 4.0 (Bruker). Database searching was performed using an in-house Mascot server, in which the MTHFR database was searched for variable modifications including oxidation (M) and phosphorylation (ST or Y) with an MS tolerance of 1.4 Da, MSMS tolerance of 0.5 Da, partials at 4, C$^{13}$ at 1 and charge states of +1, +2, +3. All putative phosphopeptide assignments were evaluated manually with an assumed false discovery rate of zero. Composite MSMS for MTHFR coverage was 92%.

**Data availability.** The crystal structures of HsMTHFR and ScMET12 have been deposited in the protein data bank (PDB) with the identifiers 6CFX and 6FNU, respectively. The full mass spectrometry phosphorylation dataset has been submitted to MassIVE with the accession number MSV000082179. All other data are available from the corresponding authors upon reasonable request.

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

## Acknowledgements

This work was supported by the Olga-Mayenfisch Stiftung (to D.S.F.), the Rare Disease Initiative Zurich, a Clinical Research Priority Program from the University of Zurich (to D. S. F. and M. R. B.) and the Swiss National Science Foundation (SNSF 31003A_156907) (to M.R.B.). The SGC is a registered charity (number 1097737) that receives funds from AbbVie, Bayer Pharma AG, Boehringer Ingelheim, Canada Foundation for Innovation, Eshelman Institute for Innovation, Genome Canada, Innovative Medicines Initiative (EU/EFPIA) [ULTRA-DD grant no. 115766], Janssen, MSD, Merck KGaA, Novartis Pharma AG, Ontario Ministry of Economic Development and Innovation, Pfizer, São Paulo Research Foundation-FAPESP, Takeda and Wellcome Trust [106169/ZZ14/Z]. We thank the Diamond Light Source for beamtime, and the staff of beamline I04-1 for assistance with data collection. We thank Merima Forny for assistance with sequencing.

## Author Contributions

D.S.F., M.R.B. and W.W.Y. conceived of the study. D.S.F., J.K. and E.R. cloned, purified and crystallized MTHFR constructs. They also performed SAM binding and dephosphorylation assays. J.K., A.O. and G.A.B. performed X-ray structure determination. G.A.B. performed and analysed the SAXS measurement. T.S. and S.L. performed and analysed the activity assays. R.C. and O.B. performed and analysed the mass spectrometry assays. D.S.F. and W.W.Y. wrote the manuscript with editing and proofreading from the other co-authors.

## Additional information

**Competing interests:** The authors declare no competing interests.

