## [Peer Review File · Nature Communications]

Reviewers' comments:

Reviewer #1 (Remarks to the Author):

In this excellent manuscript, Froese et al describe the first crystal structure of human 5,10-methylenetetrahydrofolate reductase. This is accompanied by a very thorough biochemical investigation of the functional roles of the N-terminal multiple phosphorylations as well as of the SAM-binding regulatory domain. Methylenetetrahydrofolate reductase is a textbook enzyme which is at the heart of folate metabolism. In a landmark paper published 20 years ago, Mathews, Ludwig and coworkers reported the crystal structure of the *E. coli* enzyme. The results presented in this manuscript will represent another landmark in the field because they reveal key and specific features of the more complex human protein. Above all, the X-ray analysis reveals a new folding topology for the regulatory domain which is very rare these days given that the PDB contains >100000 structures. Moreover, very well-thought and executed experiments indicate that the SAM domain affects catalysis through a long-range effect, possibly decreasing affinity for FAD. Remarkably, SAH does not have a similar effect, which is perfectly in line with the physiological and metabolic significance of such a regulatory (inhibitory) mechanism. An added value of this work is that it explains the effects of the mutations found in patients affected folate deficiencies. Of course, this study raises many more questions and more articles are likely to follow in the next years to fully clarify the molecular mechanism of enzyme regulation as well as the effect exerted by multiple phosphorylations.

I do not have major comments. The data are solid and the manuscript is very clearly written. My only concern is about the SAXS data. Chi-square values of 5-10 do not mean much. I would refrain from presenting models based on these data. I would rather limit the interpretation of the SAXS experiments to the conclusion that they fully support the notion that the enzyme is flexible and the crystal structure is rigidified by packing. I also recommend to show the plot of R_g as function of the elution volume following the guidelines given in *Acta Crystallogr D Struct Biol.* 73: 710.

Reviewer #2 (Remarks to the Author):

This manuscript focuses on human methylenetetrahydrofolate reductase (MTHFR), which catalyzes the conversion of methylenetetrahydrofolate to methyltetrahydrofolate for use in the methionine cycle. The work describes the much-anticipated crystal structure of the human enzyme with details of the well-conserved FAD-containing catalytic domain, a novel S-adenosylmethionine-binding regulatory domain, and the linker connecting them. The study combines structural, biochemical, and biophysical data to provide a basis for understanding the relationship between phosphorylation, S-adenosylmethionine sensitivity/inhibition, and conformational change.

The work has been well executed and the manuscript does an effective job of describing the results and the implications of the work. I recommend that the manuscript be accepted for publication in *Nature Communications* after addressing the following points.

1. Lines 550-561, p. 14 –Since this work is the first time that you have used the physiological forward assay with purified human MTHFR (your references all describe assays of cell lysates) and no kinetic plots are shown in the manuscript or in the Supplementary Information, please provide more details of your assays in the Materials and Methods. State the range of CH₂-THF concentrations and the saturating NADPH concentration used for the determination of the apparent K_m of CH₂-THF and the range of NADPH concentrations and the saturating CH₂-THF concentration used for the determination of the apparent K_m of NADPH.

2. Lines 224-225, p. 7, Figure 6 - Since the N-terminus of the HsMTHFR36-644 construct projects toward the dimer interface, I am curious as to whether crystallization of the full 1-656 (or 1-644) HsMTHFR was attempted. (It is stated that phosphorylation at the N-terminal could induce a

conformational change that primes the enzyme for inhibition by S-adenosylmethionine.) Was the HsMTHFR1-656 too flexible to be crystallized? In order to model the N-terminal of HsMTHFR, I think that an overlay or mention of the EcMTHFR (1ZP3) structure from Pejchal et al, 2005, which begins at residue 3, would be helpful.

3. Lines 263-311, p. 8 - I am curious why the catalytic domain of the yeast homolog MET12 (ScMET121-301) was crystallized rather than MET13, since according to Raymond, Kastanos, and Appling [Arch. Biochem. Biophys. (1999) 372, pp. 300-308], MET12 is less active than MET13 and does not demonstrate methionine auxotrophy. Was the activity of the ScMET121-301 construct tested to show that it was a functional MTHFR?

4. Line 308, Supplementary Figure 10 - It is stated here that Phe223 of EcMTHFR is replaced by Leu268 in HsMTHFR, but the alignment in Supplementary Figure 10 does not show these residues aligned.

Minor Points:

Line 78 – References for the CH3-THF-bound structure should also include 7.

Line 137, Table 1 –The kinetic parameter *k_{cat}* needs to have a lowercase *k*.

Line 215 – Should be domains, not domain.

Line 220 – Eliminate the second "in", should be "or are within two residues of".

Line 281 – Should be reducing equivalents.

Line 321 - Citations here have a different format than the others.

Line 342 – Add "as", should be "such as humans".

Lines 538, 548 – Should be Table 2, not Table 1.

Line 546 – Update to include the accession code for the yeast catalytic domain.

Table 1 – It would be helpful to include as a footnote to the table at what temperature and length of time the heat treatment was performed.

Figure 6D – The ligand should be labeled as CH3-THF not CH2-THF.

Figure 6A legend - There are 4 red arrows, but they are not labeled a-d.

Reviewer #3 (Remarks to the Author):

The present article provides the first structure of an almost full-length eukaryotic MTHFR enzyme; displaying not only the well-known N-terminal catalytic domain adopting a TIM barrel fold, but also its C-terminal regulatory domain that adopts a novel fold. Unexpectedly, this rather flexible hetero-dimer is linked via its regulatory domains that would be in close proximity with the N-terminal phosphorylated stretch (absent in the structure). Complementary bottom-up, and intact protein MS experiments show that the MTHFR can harbour concomitantly up to 10 phosphorylations on 16 different sites that have been localised in this work. Native MS further nicely confirms that the protein forms dimers and is purified with endogenous FAD and SAM. SAXS experiments support the idea of intrinsic inter-domain flexibility in solution. Enzymatic assays indicate that phosphorylation does not influence directly the activity of the enzyme but rather its binding (inhibition) to SAM.

Finally, structural details about the evolution of the catalytic site are gathered with the high resolution structure of the catalytic site from *Saccharomyces cerevisiae*.

This article is particularly well written and, even though the first 37 amino-acids that were shown to be highly phosphorylated are missing in the structure, providing structural details of this

important human enzyme is of high interest. It enables to better understand the link between the two domains and rationalises, from a structural point of view, the 70 inherited mutations found in the literature. The main conclusions align with the data and clearly show how SAM and FAD binding, together with Nter phosphorylation allosterically regulate the enzyme inhibition.

I believe that this article is of interest to a broad readership and should be considered for publication in Nature communications with editorial revisions.

Suggestions below are designed to enhance its communication and impact for the non-specialist.

Major comments:

- Up to ten concomitant phosphorylation sites were found in the 1-656 construct but the truncated one (38-644) that still contains 5 phosphosites is not phosphorylated. The authors suggest that priming by phosphorylation of the N-terminal part might be required for further phosphorylation of the other downstream residues. In order to support these findings and reach the high standards of Nature communications:

- o The full MSMS dataset should be deposited in an appropriate repository such as MassIVE or PRIDE for public access.

- o A figure representing MSMS spectra for the 16 phosphosites should be added to the supplementary material. The spectra should be clearly annotated, including b and/or y reporter ions that enabled to precisely localize the different phosphosites.

- 16 phosphosites were identified but only 9-10 were present at the same time. This result implies a strong heterogeneity in terms of combinations of phosphosites. Did the authors find any multi-phosphorylated peptide (especially on the N-terminal part)? Top-down MS could be an option to be considered in the future to investigate such heterogeneity. Could the authors please comment a bit more on this?

- For the sake of clarity, could the authors increase the fonts used in MS spectra from Fig2 and Sup. Fig2 ?

- In Sup Fig2A, residues in green are supposed to be the ones phosphorylated in the 1-656 construct but Y90/T94S394 and T451 are shown in red in Fig2A but not in green in Sup Fig2A, whereas S109 and T417 are shown in green in Sup Fig2A but not in red in Fig2A? Could the authors clarify this point ?

- Fig2B and C show very nice native MS data. I am just wondering whether the theoretical MW indicated corresponds to the SS or SH form as the 2 Da Mass difference could originate from the reduction of a SS bond?

- The question whether phosphorylation increases the binding of SAM could be answered by native MS? Has this been considered by the authors?

- Although charge state distribution is usually indicative of the folding state of a protein, in this particular case, the shift towards higher charge states for dephosphorylated MTHFR is not surprising as each phosphorylation brings negative charges so it does not necessarily corroborate a conformational change. There are many other ways to confirm this conformational change (ion mobility, limited proteolysis, hydrogen-deuterium exchange MS, SEC-MALS or even aSEC as used later for the conformational change of the catalytic domain...). FigS9 should be either removed or completed by additional experiments to support this conformational change.

- When looking at the effect of SAM/SAH binding on the catalytic domain of MTHFR with increasing lengths of the linker, it would also be interesting to show the effect of SAM/SAH binding by aSEC on the longer (38-644 & 1-656) constructs, if possible.

- For the sake of clarity, I think that the paper would benefit from an additional figure schematically summarising the main findings of the paper, or at least the proposed mechanism for the allosteric regulation of the MTHFR.

Minor comments:

In the supplementary material, line 237: "alkylation was performed by addition...". The word "performed" is missing.

Reviewer #4 (Remarks to the Author):

Froese et al report on novel crystal structures of human MTHFR, a revealing the novel fold of a SAM-binding domain unique to eukaryotes and proving insights to allosteric regulation of this enzyme by the domain. Overall, this article is well-written, and the insights are novel and the implications are of significant impact. However, technical issues need to be addressed to better support the conclusions drawn:

A. With regards to the 2.5Å crystal structure of HsMTHFR38-644, the metrics for coordinate geometries in the second chain of the asymmetric unit (B) reported in the PDB validation report are alarmingly poor for the protein backbone and Ramachandrans. Was the chain refined? It is suspected that correcting these issues will greatly decrease the gap between R work and R free. The refinement of this chain must be improved and Ramachandrans for both structures reported should be reported in Table 2.

B. There are shortcomings in the SAXS analysis and reporting that need to be resolved:

a. How were the samples prepared for analysis? The gel filtration profiles presented throughout the paper are very suggestive of polydisperse materials. SEC was used for SAXS analysis, but no described in detail are the peaks from SEC used for analysis, including what number of exposures from what regions of the elution profile were averaged together.

b. Given that SEC-SAXS was employed, it is hard to understand how $I(0)$ can be carefully compared to surmise changes in molecular volume and structure (no access to $I(0)/c$?). What does a comparison of the phosphorylated and unphosphorylated profiles using the ALS Vr Server reveal? What do the residual plots look like? What is the determined Mass by V_c in both form? What do the residuals between the two profiles look like when superposed? P_r ? What are the calculated Porod volumes?

c. Representative Guinier Plots should be shown and R_g values reported with residuals shown in the phosphorylated and unphosphorylated forms to support the conclusions made.

d. For the DAMMIF reconstructions shown, a NSD with deviation should be reported, along with Chi statistics. Were symmetry constraints employed?

e. In Fig 5 figure A, residuals should be shown to better reveal what parts of the experimental profiles do not agree with those calculated from atomic structures. Relatively, the Chi2s are relatively high and of concern.

f. The authors invoke "dynamic." What is the agreement between CORAL solutions visually and statistically when the calculation is run many times? (say 10). Is there evidence for flexibility in the experimental data from Kratky and Porod-Debye analysis? This would explain why the Chi2 in the final models remain high: no single model could explain the experimental data, and an ensemble of structures would need to be invoked to describe the solution data. This should be addressed

The authors are referred to these expert reviews with regards to reporting and analysis:

-Nat Protoc. 2014 Jul;9(7):1727-39. doi: 10.1038/nprot.2014.116. Epub 2014 Jun 26.

-Acta Crystallogr D Struct Biol. 2017 Sep 1;73(Pt 9):710-728. doi: 10.1107/S2059798317011597. Epub 2017 Aug 18.

-Protein Sci. 2010 Apr;19(4):642-57. doi: 10.1002/pro.351.

C. Mass Spectrometry is not the first choice for assigning oligomeric properties to a protein, as it does not speak to the association constant. In lieu of having SEC-MALS data, are the calculated Stokes radii from a calibrated column consistent with the different structures presented? (calculated using a program like Hydropro or SOMO). Are the gel filtration profiles consistent with varying injection concentrations? What is the running buffer? Why aren't molecular weight markers provided (can this column resolve monomer and dimer)?

Reviewers' comments:

Reviewer #1 (Remarks to the Author):

In this excellent manuscript, Froese et al describe the first crystal structure of human 5,10-methylenetetrahydrofolate reductase. This is accompanied by a very thorough biochemical investigation of the functional roles of the N-terminal multiple phosphorylations as well as of the SAM-binding regulatory domain. Methylenetetrahydrofolate reductase is a textbook enzyme which is at the heart of folate metabolism. In a landmark paper published 20 years ago, Mathews, Ludwig and coworkers reported the crystal structure of the *E. coli* enzyme. The results presented in this manuscript will represent another landmark in the field because they reveal key and specific features of the more complex human protein. Above all, the X-ray analysis reveals a new folding topology for the regulatory domain which is very rare these days given that the PDB contains >100000 structures. Moreover, very well-thought and executed experiments indicate that the SAM domain affects catalysis through a long-range effect, possibly decreasing affinity for FAD. Remarkably, SAH does not have a similar effect, which is perfectly in line with the physiological and metabolic significance of such a regulatory (inhibitory) mechanism. An added value of this work is that it explains the effects of the mutations found in patients affected folate deficiencies. Of course, this study raises many more questions and more articles are likely to follow in the next years to fully clarify the molecular mechanism of enzyme regulation as well as the effect exerted by multiple phosphorylations.

I do not have major comments. The data are solid and the manuscript is very clearly written. My only concern is about the SAXS data. Chi-square values of 5-10 do not mean much. I would refrain from presenting models based on these data. I would rather limit the interpretation of the SAXS experiments to the conclusion that they fully support the notion that the enzyme is flexible and the crystal structure is rigidified by packing. I also recommend to show the plot of R_g as function of the elution volume following the guidelines given in *Acta Crystallogr D Struct Biol.* 73:710.

We have now included a plot of R_g as a function of the elution volume in new Supplementary Figure 11 (panel a). We much appreciate the Reviewer's suggestions on revisiting our interpretation of the SAXS data:

We reasoned that the drop in chi-square values from 14.8 to 5.5, after applying CORAL modification to the structure model, indicates an improvement of model fit to the scattering data. We accept that these values remain relatively high, and expectedly so as a consequence of the protein flexibility where no single model solution can fully explain the conformational flexibility of the solution sample. We believe

there is value in describing this to illustrate the dynamics of the protein beyond what crystallography can reveal. Also, as suggested by Reviewer #4, we have provided the Residuals in Figure 5 to illustrate which parts of the experimental profiles do not agree with those calculated from atomic structures (e.g. around the $q=0.1 \text{ \AA}^{-1}$ region).

Taking on board feedback from this Reviewer and Reviewer #4, we have made alterations in the text comparing the SAXS data for phosphorylated and dephosphorylated MTHFR to limit our interpretation, as well as removing the DAMMIF model (old Figure 5b) and the I(0) comparison (old Figure 5c).

Reviewer #2 (Remarks to the Author):

This manuscript focuses on human methylenetetrahydrofolate reductase (MTHFR), which catalyzes the conversion of methylenetetrahydrofolate to methyltetrahydrofolate for use in the methionine cycle. The work describes the much-anticipated crystal structure of the human enzyme with details of the well-conserved FAD-containing catalytic domain, a novel S-adenosylmethionine-binding regulatory domain, and the linker connecting them. The study combines structural, biochemical, and biophysical data to provide a basis for understanding the relationship between phosphorylation, S-adenosylmethionine sensitivity/inhibition, and conformational change.

The work has been well executed and the manuscript does an effective job of describing the results and the implications of the work. I recommend that the manuscript be accepted for publication in Nature Communications after addressing the following points.

1. Lines 550-561, p. 14 –Since this work is the first time that you have used the physiological forward assay with purified human MTHFR (your references all describe assays of cell lysates) and no kinetic plots are shown in the manuscript or in the Supplementary Information, please provide more details of your assays in the Materials and Methods. State the range of CH₂-THF concentrations and the saturating NADPH concentration used for the determination of the apparent K_m of CH₂-THF and the range of NADPH concentrations and the saturating CH₂-THF concentration used for the determination of the apparent K_m of NADPH.

As requested by the Reviewer, we have added to the Methods section the information describing the range of CH₂-THF and NADPH concentrations used to determine the K_m of each substrate, as well as the saturating conditions of the alternate molecule in these experiments (Methods: MTHFR Assay). For transparency, we have also added to the Supplementary Information a graph depicting the Michaelis-Menten kinetics for each substrate (new Supplementary Fig. 4a and b).

2. Lines 224-225, p. 7, Figure 6 - Since the N-terminus of the HsMTHFR36-644 construct projects toward the dimer interface, I am curious as to whether crystallization of the full 1-656 (or 1-644) HsMTHFR was attempted. (It is stated that phosphorylation at the N-terminal could induce a conformational change that primes the enzyme for inhibition by S-adenosylmethionine.) Was the HsMTHFR1-656 too flexible to be crystallized? In order to model the N-terminal of HsMTHFR, I think that an overlay or mention of the EcMTHFR (1ZP3) structure from Pejchal et al, 2005, which begins at residue 3, would be helpful.

We have indeed attempted crystallizing the full-length HsMTHFR protein (aa1-656), although our intensive attempts were not successful. This could be due to the intrinsic flexibility of the N-terminus, or the potential heterogeneity in the phosphorylation states.

We thank the Reviewer for suggesting to compare the N-terminus of available MTHFR catalytic domain structures, including PDB 1ZP3. We have now included in new Supplementary Fig. 14 a structural overlay of MTHFR catalytic domains from human, yeast, *T.thermophilus* (PDB 3APT),

H. influenzae (5UME), and *E. coli* (1ZP3). This figure shows that the first helix of the catalytic domain ($\alpha 1$) is observed in two different orientations among the five structures. It is of note that human MTHFR contains the 38-aa phosphorylation region immediately N-terminal to the catalytic domain, not present in the biological sequence of the other structural orthologues. Therefore, the structural overlay in Supplementary Fig. 14 will not allow modelling of this N-terminal phosphorylation region – at best, the proximity of catalytic domain helix $\alpha 1$ towards the regulatory domain in HsMTHFR structure implies that the N-terminal phosphorylation preceding helix $\alpha 1$ would also be strategically posed near the regulatory domain.

To describe the above, in the Results section “Subtle features provide for eukaryotic NADPH specificity” paragraph 1, we have added: “*Additionally, the first helix of the catalytic domain ($\alpha 1$) is observed in different orientations among these structures (Supplementary Fig. 14). There is sequence divergence of helix $\alpha 1$ among prokaryotes, lower and higher eukaryotes (Supplementary Fig. 15). In HsMTHFR (which contains, in its biological sequence but not present in the crystallized construct, the serine-rich phosphorylation region N-terminal to the catalytic domain), ScMET12 and *T. thermophilus* MTHFR, this helix $\alpha 1$ is projected towards the interface between catalytic and regulator domains.*”

3. Lines 263-311, p. 8 - I am curious why the catalytic domain of the yeast homolog MET12 (ScMET121-301) was crystallized rather than MET13, since according to Raymond, Kastanos, and Appling [Arch. Biochem. Biophys. (1999) 372, pp. 300-308], MET12 is less active than MET13 and does not demonstrate methionine auxotrophy. Was the activity of the ScMET121-301 construct tested to show that it was a functional MTHFR?

MET12 was crystallized instead of MET13 because following testing of a nested set of constructs with varying boundaries we were able to produce purifiable recombinant protein from the catalytic domain of MET12 but not MET13. Using this catalytic domain we were then able to produce diffracting crystals. By contrast, we were able to express and purify the full-length protein and the regulatory domain only of both MET12 and MET13. Nevertheless, crystallization attempts of these constructs failed.

The activity of MET12₁₋₃₀₁ was not tested, nevertheless we expect it to be functional for the following reasons:

1) In the paper the Reviewer has cited (Raymond et al. Arch Biochem Biophys 1999), the authors have demonstrated activity of MET12 from *E. coli* expression (MET12 activity: 7.4 nmmol/mg protein/hour).

2) MET12 contains all catalytic residues harboured by MET13 and HsMTHFR, as illustrated by a sequence alignment between these proteins (Figure below). Consistent with a functional catalytic domain, each subunit of *as purified* MET12₁₋₃₀₁ contained an FAD molecule which, in the crystal structure, perfectly aligns (RMSD: 0Å) with FAD bound to HsMTHFR and EcMTHFR.

3) The major sequence difference between MET12 and MET13 or HsMTHFR is an extended insertion in the linker region following the catalytic domain (Figure below). Given the multiple contacts we found between the linker region and the catalytic domain and the regulatory domain in our HsMTHFR structure, this extended linker may affect overall activity or modulation of the activity by SAM, and may explain the decreased activity of the full-length MET12 protein compared to MET13.

4) Finally, structural superimposition of the catalytic domain of MET12 with HsMTHFR and EcMTHFR as shown in Fig. 6 of our paper demonstrates the strong conservation of the overall catalytic structure. This structural conservation is the major point for which we use this structure, and remains true whether this domain *in vivo* has very high or low activity.

4. Line 308, Supplementary Figure 10 - It is stated here that Phe223 of EcMTHFR is replaced by Leu268 in HsMTHFR, but the alignment in Supplementary Figure 10 does not show these residues aligned.

The Reviewer is correct that in Supplementary Fig. 13 (former Supplementary Fig. 10) EcMTHFR Phe223 aligns with Gln267 of HsMTHFR and Ala229 of ScMET12. We have corrected this in the manuscript.

Minor Points:

- Line 78 – References for the CH3-THF-bound structure should also include 7.
- Line 137, Table 1 –The kinetic parameter kcat needs to have a lowercase k.
- Line 215 – Should be domains, not domain.
- Line 220 – Eliminate the second "in", should be "or are within two residues of".
- Line 281 – Should be reducing equivalents.
- Line 321 - Citations here have a different format than the others.
- Line 342 – Add "as", should be "such as humans".
- Lines 538, 548 – Should be Table 2, not Table 1.
- Line 546 – Update to include the accession code for the yeast catalytic domain.
- Table 1 – It would be helpful to include as a footnote to the table at what temperature and length of time the heat treatment was performed.
- Figure 6D – The ligand should be labeled as CH3-THF not CH2-THF.
- Figure 6A legend - There are 4 red arrows, but they are not labeled a-d.

We have corrected these mistakes throughout the manuscript.

Reviewer #3 (Remarks to the Author):

The present article provides the first structure of an almost full-length eukaryotic MTHFR enzyme; displaying not only the well-known N-terminal catalytic domain adopting a TIM barrel fold, but also its C-terminal regulatory domain that adopts a novel fold. Unexpectedly, this rather flexible hetero-dimer is linked via its regulatory domains that would be in close proximity with the N-terminal phosphorylated stretch (absent in the structure). Complementary bottom-up, and intact protein MS experiments show that the MTHFR can harbour concomitantly up to 10 phosphorylations on 16 different sites that have been localised in this work. Native MS further nicely confirms that the protein forms dimers and is purified with endogenous FAD and SAM. SAXS experiments support the idea of intrinsic inter-domain flexibility in solution. Enzymatic assays indicate that phosphorylation does not influence directly the

activity of the enzyme but rather its binding (inhibition) to SAM.

Finally, structural details about the evolution of the catalytic site are gathered with the high resolution structure of the catalytic site from *Saccharomyces cerevisiae*.

This article is particularly well written and, even though the first 37 amino-acids that were shown to be highly phosphorylated are missing in the structure, providing structural details of this important human enzyme is of high interest. It enables to better understand the link between the two domains and rationalises, from a structural point of view, the 70 inherited mutations found in the literature. The main conclusions align with the data and clearly show how SAM and FAD binding, together with Nter phosphorylation allosterically regulate the enzyme inhibition.

I believe that this article is of interest to a broad readership and should be considered for publication in Nature communications with editorial revisions.

Suggestions below are designed to enhance its communication and impact for the non-specialist.

Major comments:

- Up to ten concomitant phosphorylation sites were found in the 1-656 construct but the truncated one (38-644) that still contains 5 phosphosites is not phosphorylated. The authors suggest that priming by phosphorylation of the N-terminal part might be required for further phosphorylation of the other downstream residues. In order to support these findings and reach the high standards of Nature communications:

- o The full MSMS dataset should be deposited in an appropriate repository such as MassIVE or PRIDE for public access.

We thank the Reviewer for the excellent suggestion - and have deposited the full MS/MS dataset into MassIVE, with accession ID: MSV000082179 (MTHR_HUMAN phosphorylation mapping MSMS dataset).

- o A figure representing MSMS spectra for the 16 phosphosites should be added to the supplementary material. The spectra should be clearly annotated, including b and/or y reporter ions that enabled to precisely localize the different phosphosites.

As suggested by the Reviewer, we have included new Supplementary Fig. 1 displaying the phosphomap, including the b and y reporter ions for each phosphosite.

- 16 phosphosites were identified but only 9-10 were present at the same time. This result implies a strong heterogeneity in terms of combinations of phosphosites. Did the authors find any multi-phosphorylated peptide (especially on the N-terminal part)? Top-down MS could be an option to be considered in the future to investigate such heterogeneity. Could the authors please comment a bit more on this?

Yes, we did observe multiply phosphorylated peptides in our MSMS dataset (see new Supplementary Fig. 1) but did not quantify them using our methods. We agree with the Reviewer that quantitation of MTHFR proteoforms would be desirable to assess whether phosphosites occur in combinations, and would merit future studies since this is non-trivial.

Top-down fragmentation, which uses electron transfer dissociation to generate N- or C-terminal fragments without neutral loss of the phosphate, is currently the preferred method for proteoform quantitation. However, to our knowledge this works best on proteins < 30 kDa in size (MTHFR monomer > 70 kDa) and different proteoforms with the same m/z can not be resolved in this way. So proteoform quantitation for MTHFR will be a difficult task.

We have added to the manuscript in the Discussion: “While 16 phosphorylation sites were identified, intact mass analysis of MTHFR showed that only 9-10 phosphorylations were present at a time, implying a high degree of heterogeneity. While multiply phosphorylated peptides were observed, they were not quantified using our methods. Identification of MTHFR proteoforms to assess whether phosphosites occur in combinations, perhaps using top-down MS³⁵ would be very interesting, and may constitute the basis of future studies.”

- For the sake of clarity, could the authors increase the fonts used in MS spectra from Fig2 and Sup. Fig2?

We have increased the fonts in this figure where possible.

- In Sup Fig2A, residues in green are supposed to be the ones phosphorylated in the 1-656 construct but Y90/T94S394 and T451 are shown in red in Fig2A but not in green in Sup Fig2A, whereas S109 and T417 are shown in green in Sup Fig2A but not in red in Fig2A? Could the authors clarify this point?

The assertions the Reviewer has made are correct, and the confusion the Reviewer had with former Supplementary Fig. 2a (current Supplementary Fig. 3a) was due to an unfortunate mix-up of this figure with the submitted manuscript. To avoid confusion, we no longer specifically colour these residues for *HsMTHFR*₃₈₋₆₄₄ in the supplementary figure. Instead, we colour them only based on mass spec coverage. The phosphorylated residues of *HsMTHFR*₁₋₆₅₆ remain correctly coloured red in Fig. 2a.

- Fig2B and C show very nice native MS data. I am just wondering whether the theoretical MW indicated corresponds to the SS or SH form as the 2 Da Mass difference could originate from the reduction of a SS bond?

This is a very interesting suggestion. Unfortunately, the theoretical mass provided in the manuscript is assuming all cysteines are reduced. Therefore, the extra hydrogens provided by a reduced disulfide bond cannot in this instance explain the 3 extra amu found in the monomeric protein form.

- The question whether phosphorylation increases the binding of SAM could be answered by native MS? Has this been considered by the authors?

On the Reviewer's suggestion we have tried to assess SAM binding by native MS. Dimeric *HsMTHFR*₁₋₆₅₆ when purified is phosphorylated and bound to 0, 1 or 2 SAM molecules (Fig. 2E) and is therefore not fully saturated with SAM. In order to test whether we can detect saturation of SAM binding, the first step towards measuring SAM affinity, we incubated this sample with 300 μM SAM at room temperature for 30 minutes, a concentration high enough to fully inhibit the enzyme as judged by our activity assay (Fig. 3). Nevertheless following this treatment, while the majority of the protein still formed intact dimer (upper panel of figure below), deconvolution of the spectrum, which was of poor quality (lower panel of figure below), still demonstrated 3 distinct peaks, consistent with 0, 1 or 2 SAM molecules bound per dimer. This suggests that detection of SAM saturation, and subsequently SAM affinity, will take a different, more refined approach, and should be considered for future experiments. It should also be noted that these future experiments should be performed in tandem with experiments to determine whether phosphosites occur in combination, as described above, to see if specific phosphosite combinations affect SAM binding and/or inhibition.

Upper panel: Charge distribution analysis of *HsMTHFR*₁₋₆₅₆ following native MS

Lower panel: Zoom in on dimeric *HsMTHFR*₁₋₆₅₆ following deconvolution to show 3 distinct peaks, likely corresponding to 0, 1 and 2 SAM molecules bound.

- Although charge state distribution is usually indicative of the folding state of a protein, in this particular case, the shift towards higher charge states for dephosphorylated MTHFR is not surprising as each phosphorylation brings negative charges so it does not necessarily corroborate a conformational change.

There are many other ways to confirm this conformational change (ion mobility, limited proteolysis, hydrogen-deuterium exchange MS, SEC-MALS or even aSEC as used later for the conformational change of the catalytic domain...).

FigS9 should be either removed or completed by additional experiments to support this conformational change.

We beg to differ with the Reviewer's assertion that dephosphorylation proportionately changes the charge state distribution due to the loss of negative charges. In a published example (Kaltashov *et al.*

2013, *Protein Sci.* 22, 530-544), the unfolding of Pepsin, an acidic protein which has only four basic residues, leads to the generation of protein ions that accommodate over 31 protons; while the extent of multiple charging of pepsin polyanions in negative ion ESI MS is very similar to the multiply protonated species in positive ion ESI MS, despite the 10-fold excess of acidic residues. In our lab, we also have the unpublished observation of an increased protonated state following increasing phosphorylation of Aurora B kinase, the opposite finding to what we observed for HsMTHFR. These data suggest that the physical dimension of the protein ultimately dictates the extent of multiple charging in ESI MS, and not the innate charges. This is directly supported by Kaltashov & Mohimen (2005, *Anal. Chem.* 77, 5370-5379), who demonstrated that native protein surface area is the sole determinant of charge state.

Our data indicates a conformational change equivalent to a 0.5% change in radius. This is beyond the resolving power of ion mobility MS, limited proteolysis, hydrogen-deuterium exchange or SEC-MALS. It is also beyond the power of analytical SEC, consistent with the lack of change observed for HsMTHFR₁₋₆₅₆ and HsMTHFR₃₈₋₆₄₄ (new Supplementary Fig. 21). However, it is consistent with the small differences identified between phosphorylated and dephosphorylated HsMTHFR₁₋₆₅₆ using the more sensitive SAXS technique (Fig. 5, new Supplementary Figure 11, new Supplementary Table 1).

We therefore prefer to leave this figure within the supplementary material of our manuscript, but as mentioned above, with severely reduced interpretation.

- When looking at the effect of SAM/SAH binding on the catalytic domain of MTHFR with increasing lengths of the linker, it would also be interesting to show the effect of SAM/SAH binding by aSEC on the longer (38-644 & 1-656) constructs, if possible.

We have performed the experiments the Reviewer has requested. This experiment supports our original conclusion that SAM and SAH binding has no effect on the oligomeric state of the full-length (HsMTHFR₁₋₆₅₆) or near full-length (HsMTHFR₃₈₋₆₄₄) protein, as neither protein showed a change in stokes radius upon ligand addition.

We have appended these results as new Supplementary Fig. 21 and added in the text “*Exposure of MTHFR₃₄₈₋₆₅₆ to either SAH or SAM resulted in shifts of elution volume (V_e) compared to as purified (apo-) protein, in contrast to HsMTHFR₁₋₆₅₆ and HsMTHFR₃₈₋₆₄₄, which did not show changes in V_e despite SAH or SAM binding (Supplementary Fig. 21).*”

- For the sake of clarity, I think that the paper would benefit from an additional figure schematically summarising the main findings of the paper, or at least the proposed mechanism for the allosteric regulation of the MTHFR.

We thank the Reviewer for the suggestion, and have included a schematic highlighting mechanistic features arising from this work, as new Supplementary Fig. 22.

It is referred to in the Discussion, first paragraph. “*Now, our structure-guided study has provided 2 major discoveries in this area: (1) identification of an extensive linker region involved in both SAM-binding and purveying the binding signal to inhibit catalysis by conformational change; and (2) demonstration of the concerted effects of phosphorylation and SAM binding, individually mediated by regions more than 300 amino acids apart (Supplementary Fig. 22).*”

Minor comments:

In the supplementary material, line 237: “alkylation was performed by addition...”. The word “performed” is missing.

This mistake has been corrected.

Reviewer #4 (Remarks to the Author):

Froese et al report on novel crystal structures of human MTHFR, revealing the novel fold of a SAM-binding domain unique to eukaryotes and providing insights into allosteric regulation of this enzyme by the domain. Overall, this article is well-written, and the insights are novel and the implications are of significant impact. However, technical issues need to be addressed to better support the conclusions drawn:

A. With regards to the 2.5Å crystal structure of HsMTHFR₃₈₋₆₄₄, the metrics for coordinate geometries in the second chain of the asymmetric unit (B) reported in the PDB validation report are alarmingly poor for the protein backbone and Ramachandrans. Was the chain refined? It is suspected that correcting these issues will greatly decrease the gap between R_{work} and R_{free} . The refinement of this chain must be improved and Ramachandrans for both structures reported should be reported in Table 2.

As described in the third paragraph of the Results section “MTHFR forms an asymmetric dimer with inter-domain flexibility”, we observed a high degree of disorder in much of chain B catalytic domain, to the extent that only main-chain atoms were built for residues 40-58, 129-134 and 155-342 in chain B. In response to the Reviewer’s request, we have now further refined chain B, significantly improving its backbone geometry as well as Ramachandran stereochemistry. The current model has 99.8% residues in all allowed regions. Additionally during the refinement, the gap between R_{work} and R_{free} is also moderately narrowed (from 4.7% to 3.7%). We will be replacing the PDB entry 6FCX with the refined model, and have also included the Ramachandran statistics in the Methods section (as per Journal requirement).

B. There are shortcomings in the SAXS analysis and reporting that need to be resolved:

a. How were the samples prepared for analysis? The gel filtration profiles presented throughout the paper are very suggestive of polydisperse materials. SEC was used for SAXS analysis, but no details are described in detail of the peaks from SEC used for analysis, including what number of exposures from what regions of the elution profile were averaged together.

The full-length MTHFR samples for SAXS measurements were either *as purified* to represent the phosphorylated sample (described in Methods, ‘*Recombinant production of MTHFR*’), or further treated with CIP. We have now provided the CIP dephosphorylation procedure in the Methods text.

During the SEC-SAXS runs, we chose frames from the peak regions where the calculated R_g values are constant. To illustrate this point, we have now provided the plot of R_g as a function of the elution volume in new Supplementary Fig. 11, panel a. As stated in the legend, the frames used for further analysis are 305-315, 305-320 and 300-315 respectively for MTHFR₃₈₋₆₄₄, CIP-treated MTHFR₁₋₆₅₆, and *as purified* MTHFR₁₋₆₅₆. Exposure of 3 seconds per frame was used.

b. Given that SEC-SAXS was employed, it is hard to understand how $I(0)$ can be carefully compared to surmise changes in molecular volume and structure (no access to $I(0)/c$). What does a comparison of the phosphorylated and unphosphorylated profiles using the ALS Vr Server reveal? What do the residual plots look like? What is the determined Mass by V_c in both forms? What do the residuals between the two profiles look like when superposed? P_r ? What are the calculated Porod volumes?

We appreciated the concern of the Reviewer in our interpretation of $I(0)$ for changes in molecular dimensions, and have therefore opted to remove description of the $I(0)$ data (i.e. old Figure 5c).

As requested by the Reviewer, we have used the ALS Vr Server which did not reveal any informative difference between phosphorylated and unphosphorylated profiles. We have also determined $P(r)$ function and Porod volumes between the two samples. With these parameters (see table below, excerpted from new Supplementary Table 1), we observe small differences between the CIP-treated

(dephosphorylated) and as *purified* (phosphorylated) samples. We did not calculate residuals of the two profiles; to the best of our knowledge, residuals are a statistical treatment of the difference between an observed value and the predicted values. In our case, the two experimental profiles (phosphorylated and unphosphorylated) are therefore deemed not subtractable.

	MTHFR₁₋₆₅₆ CIP-treated	MTHFR₁₋₆₅₆ as-purified
Structural parameters		
Rg (Å, from Guinier)	40.8 ± 0.5	41.3 ± 0.5
Volume (Å ³ , from Porod)	220	254
Mr determination (kDa)*		
Estimated from Porod volume	137	158
Estimated from volume-of-correlation (Vc)	140	150

*Mr calculated from sequence: 150.7 kDa

To describe these comparisons, we have included a supplementary table and written the following in the text: “*The experimental scattering curves for as purified and CIP-treated HsMTHFR₁₋₆₅₆ gave rise to slightly different profiles and derived parameters (Supplementary Fig. 11, Supplementary Table 1), although both protein forms are consistent with a dimeric configuration.*”

c. Representative Guinier Plots should be shown and Rg values reported with residuals shown in the phosphorylated and unphosphorylated forms to support the conclusions made.

We have included the Guinier Plots with reported Rgs and residuals for all the measured SAXS profiles, which are presented in new Supplementary Figure 11, panel b.

d. For the DAMMIF reconstructions shown, a NSD with deviation should be reported, along with Chi statistics. Were symmetry constraints employed?

In light of Reviewer #1’s suggestions to refrain from presenting SAXS models, we have decided to remove the shape reconstruction by DAMMIF (former Fig. 5b) from the manuscript. We believe the DAMMIF model does not add further to the x-ray structure, and removing it does not impact on other SAXS data.

e. In Fig 5 figure A, residuals should be shown to better reveal what parts of the experimental profiles do not agree with those calculated from atomic structures. Relatively, the Chi2s are relatively high and of concern.

We have altered Figure 5a to include the residuals in the bottom panel of the figure, which showed differences between the experimental and calculated profiles around the $q=0.1 \text{ \AA}^{-1}$ region. We agree with the Reviewer’s view (see f below) that the high Chi2 is likely a consequence of the fact that no single model can fully explain the experimental data because of the intrinsic flexibility of the molecule. As described in our response to Reviewer #1, we believe there is value in describing these chi-square values, to illustrate the dynamics of the protein beyond what crystallography can reveal.

f. The authors invoke “dynamic.” What is the agreement between CORAL solutions visually and statistically when the calculation is run many times? (say 10). Is there evidence for flexibility in the experimental data from Kratky and Porod-Debye analysis? This would explain why the Chi2 in the final models remain high: no single model could explain the experimental data, and an ensemble of structures would need to be invoked to describe the solution data. This should be addressed

To improve statistical confidence, we have now run CORAL 10 times and the average Chi2 of all the CORAL runs is 2.95 +/- 0.18, suggesting that the CORAL solutions are statistically consistent. The Kratky and Porod-Debye analysis did not show a classical profile of flexible samples. We attribute this to the fact that the suggested analyses are more sensitive to detection of unfolded or partially unfolded proteins. Therefore, Kratky and Porod-Debye analysis might not be sensitive enough to capture the relatively subtle rearrangement undergone by the catalytic and regulatory domains of MTHFR.

The authors are referred to these expert reviews with regards to reporting and analysis:

-Nat Protoc. 2014 Jul;9(7):1727-39. doi: 10.1038/nprot.2014.116. Epub 2014 Jun 26.

-Acta Crystallogr D Struct Biol. 2017 Sep 1;73(Pt 9):710-728. doi: 10.1107/S2059798317011597. Epub 2017 Aug 18.

-Protein Sci. 2010 Apr;19(4):642-57. doi: 10.1002/pro.351.

C. Mass Spectrometry is not the first choice for assigning oligomeric properties to a protein, as it does not speak to the association constant. In lieu of having SEC-MALS data, are the calculated Stokes radii from a calibrated column consistent with the different structures presented? (calculated using a program like Hydropro or SOMO). Are the gel filtration profiles consistent with varying injection concentrations? What is the running buffer? Why aren't molecular weight markers provided (can this column resolve monomer and dimer)?

Although not explicitly stated, the experiments to which we believe the Reviewer is referring concern the N-terminally truncated constructs of MTHFR regulatory domain (Fig. 7, Supplementary Fig. 20), as at the time of submission these are the only SEC figures presented.

As described in the Methods section of the referenced paper (Froese *et al.*, 2012 *Biochem.* 5, 5083-5090) the conditions and column used in our SEC experiments were as follows. *Analytical gel filtration was performed on a Superdex 200 HiLoad 10/30 column (GE Healthcare) pre-equilibrated with 10 mM HEPES pH 7.5, 150 mM NaCl and 5% glycerol. The column was calibrated using carbonic anhydrase (29 kDa), bovine serum albumin (66 kDa), alcohol dehydrogenase (150 kDa), and apoferritin (443 kDa) as protein standards.* For clarity, the italicized information is now clearly stated in the Methods section of this manuscript. This column can certainly distinguish between the monomeric and dimeric forms of the protein (resolves between 10-600 kDa), though as discussed below, assignment of oligomeric state to our truncated constructs is not important.

The purpose of the N-terminal truncation experiments was to identify changes in solution behaviour of the Regulatory domain-alone proteins with progressively smaller linker sequences (constituting aa 348-656, 357-656 or 377-656) following their incubation with SAM and SAH. In this way, it was not important if these truncated proteins, which are essentially artificial constructs, turned out to be monomeric, dimeric or higher order species. What was important was that the elution volume changed upon addition of SAM and/or SAH for the larger constructs, a change which disappeared with increasing N-terminal truncation. Our conclusion was therefore, that following SAM/SAH binding, the elution volume of the protein must have changed, indicating a change of state due to re-arrangement of the protein that is dependent on certain amino acids that are present in the longer constructs but absent in the shorter constructs (in this case amino acids 357-377), based upon our observation that the longer constructs changed elution volume upon SAM/SAH addition but the shorter constructs did not.

We have decided to remove the sentences: "...which we interpreted as changes in the overall protein conformation, rather than changes in oligomeric states. Our assumption is based on the native mass spectrometry data (Fig. 2C; Supplementary Fig. 2B) showing that SAM and SAH do not alter the oligomeric states observed for the HsMTHFR₁₋₆₅₆ and HsMTHFR₃₈₋₆₄₄ proteins." from the Results, in the section "The linker mediates SAM-dependent conformational change", cognizant that the intent of

these sentences to support our interpretation may be confusing to the Reviewer/reader. Instead, at the conclusion of this paragraph we state “*Therefore, we conclude that residues within 357-377 must contribute to change of protein state upon SAM binding, which we interpret as a change in conformation.*” making clear that this is our interpretation although others may exist.

Finally, as discussed in Sharon and Robinson (2007, *Ann. Rev. Biochem.* 76, 167-193), native mass spectrometry is actually an ideal tool for assigning oligomeric state. However, in our case the dimeric state of the Full-Length (*HsMTHFR*₁₋₆₅₆) and/or near Full-Length (*HsMTHFR*₃₈₋₆₄₄) were also identified by SAXS (new Supplementary Table 1) and in the crystal structure (Fig. 4).